# An anti-influenza combined therapy assessed by single cell RNA-sequencing

Chiara Medaglia [1,6✉], Ilya Kolpakov[2,6], Arnaud Charles-Antoine Zwygart[1], Yong Zhu[3], Samuel Constant[4], Song Huang[4], Valeria Cagno[5], Emmanouil T. Dermitzakis [2], Francesco Stellacci [3], Ioannis Xenarios[2] & Caroline Tapparel [1✉]

Influenza makes millions of people ill every year, placing a large burden on the healthcare system and the economy. To develop a treatment against influenza, we combined virucidal sialylated cyclodextrins with interferon lambda and demonstrated, in human airway epithelia, that the two compounds inhibit the replication of a clinical H1N1 strain more efficiently when administered together rather than alone. We investigated the mechanism of action of the combined treatment by single cell RNA-sequencing analysis and found that both the single and combined treatments impair viral replication to different extents across distinct epithelial cell types. We showed that each cell type comprises multiple sub-types, whose proportions are altered by H1N1 infection, and assessed the ability of the treatments to restore them. To the best of our knowledge this is the first study investigating the effectiveness of an antiviral therapy against influenza virus by single cell transcriptomic studies.

[1] Department of Microbiology and Molecular Medicine, University of Geneva, 1206 Geneva, Switzerland. [2] Health 2030 Genome Center, 1202 Geneva, Switzerland. [3] Insitute of Materials, Ecole Polytechnique Fédérale de Lausanne, 1015 Lausanne, Switzerland. [4] Epithelix Sas, 1228 Geneva, Switzerland. [5] Faculty of Biology and Medicine, Université de Lausanne, 1011 Lausanne, Switzerland. [6] These authors contributed equally: Chiara Medaglia, Ilya Kolpakov. ✉email: chiara.medaglia@fht.org; caroline.tapparel@unige.ch

nfluenza is a highly contagious respiratory infection that accounts every year for about ~3–5 million cases of severe illness and up to 650,000 deaths[1]. More than a century after the Spanish pandemic, the health systems are still struggling to cope with seasonal influenza, something that bodes poorly in the event of a novel pandemic. Influenza is caused, in humans, by influenza A (IAV) and influenza B (IBV) viruses. Although the latter are almost exclusively found in the human population, IAVs emerge from a huge zoonotic reservoir[2]. In a process called antigenic drift, IAVs rapidly acquire adaptive mutations allowing them not only to evade the host immune response but also to neutralize annual attempts to generate effective vaccines[3]. As a consequence, seasonal epidemics endanger every year children, elderly people, pregnant women, and people of any age with comorbid illnesses[4]. In addition, due to their ability to cross the species barrier, IAVs pose a high pandemic risk. The arrangement of the viral genome on multiple RNA segments allows for the exchange of genetic material between different viral strains which co-infect the same host, giving rise to novel gene-reassorted variants. This process, when accompanied by the expression of new surface glycoproteins, is named antigenic shift, as it results in the emergence of strains that infect immunologically naive humans and cause potentially pandemic outbreaks[5]. Lastly, when the reassortant viruses possess new virulence factors, they can be associated with increased pathogenicity.

Influenza virus (IV) is enveloped, with a negative single strand RNA genome. The viral protein hemagglutinin (HA) of human IV binds preferentially α2,6-linked sialic acid (Sia) moieties located on the surface of the host cell, thus triggering viral entry through clathrin-mediated endocytosis[6]. Upon entering a new host, IV establishes infection in the epithelial cells lining the upper airways[7]. When the infection stays restricted to this region of the respiratory tract, it causes rather mild disease. But if it spreads to the lungs, it can cause viral pneumonia, with progression to acute respiratory distress syndrome (ARDS) and death from respiratory failure[8]. IAV disrupts the functions of the respiratory barrier by inducing epithelial cell death via intrinsic viral pathogenicity, or through a robust immune response[9]. This alteration leads to exposure of new attachment sites for bacteria[10], thus making the host more vulnerable to secondary infections by other pathogens, which significantly contribute to the morbidity of influenza[11].

Annual vaccination is the cornerstone of prevention against IVs. However, the vaccine has to be adapted yearly and does not always match with circulating strains. This is further complicated by the co-circulation of different IV types and different IAV subtypes[12]. In the 2017–2018 United States season, vaccine effectiveness was estimated to be only ~25% against influenza A subtype H3N2 viruses, which, however, comprised ~69% of infections[13]. Antivirals represent an important second line of defense against IV, but all the currently available drugs are only efficient if taken at the early stages of the disease. Moreover, they inevitably exert selective pressure on the virus, which causes the appearance of drug-resistant variants[14–16]. It results that there is an unmet need to develop additional therapies against IV.

Several studies indicate IFN λ as a promising therapeutic candidate for controlling influenza and other viral respiratory diseases[17,18]. The family of IFN λ (alias IFN type III) comprises IFN λ1, IFN λ2, and IFN λ3 (also known as IL-29, IL-28A, and IL-28B, respectively), and the recently identified IFN λ4[19]. Like IFN type I, IFN λ acts both in an autocrine and paracrine fashion, inducing an antiviral state through the expression of interferon-stimulated genes (ISGs), that inhibit viral replication at multiple steps[20]. The antiviral state induced by IFN λ is localized to the mucosal surfaces, as the expression of its receptor is mostly restricted to the epithelial cells of the gut and the respiratory

tract[21]. Indeed, immune cells are largely unresponsive to IFN λ[21,22]. Thus, while IFN type I targets nearly all immune cells, creating massive inflammation that may further weaken the host[23], IFN λ only acts at the epithelial barriers and on a few innate immune cells, without causing immunopathology[18,24]. These properties suggest IFN λ as a treatment of choice against acute viral infections, such as influenza, with higher tolerability than IFN type I. IFN λ plays a critical early role, not shared by IFN type I, in the protection of the lung following IV infection[25–28], and several in vivo studies show that it also exerts variable degrees of antiviral activity against both IAV and IBV strains[29]. It has been reported that, in B6.A2G-MX1 mice infected with H1N1 IAV, IFN λ intranasal administration prevents viral spread from the upper to the lower airway, without noxious inflammatory side effects[26,30]. Importantly, human pegylated IFN λ1 passed both phase I and phase II clinical trials for hepatitis C treatment, displaying an attractive pharmacological profile[31,32].

Combination therapy is considered a valuable approach to provide greater clinical benefit, especially to those at risk of severe disease. Combining drugs targeting different mechanisms of viral replication may increase the success rate of the treatment[33,34], as also demonstrated in our previous work, showing that IFN λ1 co-administration delays the emergence of H1N1 IAV resistance to oseltamivir[35]. We recently developed 6'SLN-CD [heptakis-(6-deoxy-6-thioundec)-beta-cyclodextrin grafted with 6'SLN(Neu5Ac-a-(2-6)-Gal-b-(1-4)-GlcNAc;6'-N-Acetylneuraminyl-N-acetyllactosamine], a nontoxic anti-influenza antiviral designed to target and irreversibly inactivate extracellular IV particles, preventing their entrance into the host cell. 6'SLN-CD significantly decreases IAV replication in both ex vivo and in vivo models of infection[36]. However, 6'SLN-CD targets the globular head of IV HA, which undergoes constant antigenic drift, thus posing a concrete problem of resistance emergence [14a]. In this work, we chose to combine human IFN λ1, the host frontline defense against IAV, with 6'SLN-CD, in order to increase its effectiveness and lower the chances of antiviral resistance. The two compounds hinder viral replication on different fronts: IFN λ1 boosts the host innate response while 6'SLN-CD traps and inactivates newly formed virions. To mimic the in vivo environment, we assessed the combinatorial effect of the compounds in 3D human airway epithelia (HAE) reconstituted at the air-liquid interface[11,37] and showed that IFN λ1 enhances 6'SLN-CD antiviral activity. HAE perfectly mimics both the pseudostratified architecture of the human respiratory epithelium, composed of basal, ciliated, and secretory cells, and its defense mechanisms. In addition, they allow the use of clinical viral specimens, thus preserving their original pathogenicity and biological characteristics, which are inevitably lost upon repeated passages in cell lines[37–40].

As host cellular heterogeneity strongly impacts virus-host interplay and is mirrored in response to antiviral treatments[41,42], we investigated the mechanism of action of IFN λ1 plus 6'SLN-CD by single cell RNA-sequencing (scRNA-seq). This approach allowed us to trace the landscape of the modifications through which individual cells respond to IAV infection and to the treatments. We found that in different epithelial cell types, both the individual and the combined antivirals hinder viral replication to different extents, depending on the permissiveness of the cells to H1N1. We also showed that each basal, secretory, and ciliated cells comprise multiple subclusters, whose proportions are altered by the infection. Surprisingly even though in each cell type, the antivirals reduced viral replication synergistically, they were not able to restore the changes in cell subcluster composition in a similar manner. Lastly, in the absence of infection, IFN λ1 + 6'SLN-CD did not alter the proportions of the main epithelial cell types, further supporting the therapeutic potential of

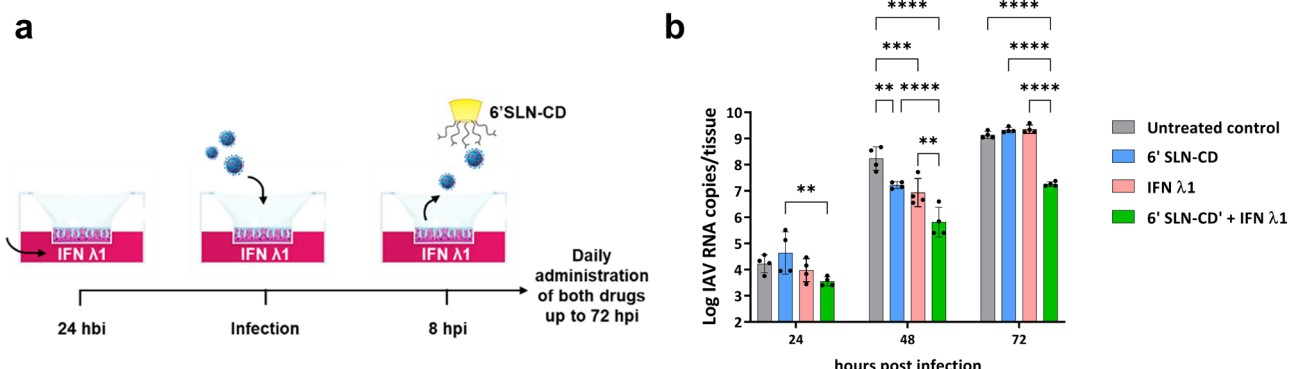

**Fig. 1 Combinatorial effect of 6'SLN-CD + IFN λ1 in HAE. a** Schematics of 6'SLN-CD and IFN λ1 (60 μg and 5.5 ng per tissue, respectively) combined administration. **b** Bar plot showing the kinetic of IAV replication in HAE treated with 6'SLN-CD only, or with IFN λ1 only, or with both compounds according to **a**. The results represent mean and standard deviation (calculated with the *t*-test analysis) of two independent experiments conducted in duplicate in HAE developed from a pool of donors and infected with 10³ RNA copies of clinical A/Switzerland/3076/2016 H1N1 (0 h corresponds to the time of viral inoculation). Viral replication was assessed measuring the apical release of IAV by RT-qPCR. **p ≤ 0.01; ***p ≤ 0.001; ****p ≤ 0.0001. HAE = human airway epithelia.

the formulation. The findings presented in this work pave the way for future in vivo experiments, to better assess the efficacy of IFN λ1 + 6'SLN-CD treatment against influenza. To the best of our knowledge, this is the first study investigating the effectiveness of antiviral therapy against influenza by scRNA-seq.

## Results

**IFN λ1 and 6'SLN-CD display synergistic activity against H1N1 IAV ex vivo.** We optimized an IFN λ1/6'SLN-CD formulation to inhibit IAV in ex vivo 3D HAE. The tissues were infected with a clinical A/Switzerland/3076/2016 H1N1 strain that has not been passaged in cell lines, to exclude any in vitro adaptation bias. First, we determined the best administration mode for the individual treatments. While 6'SLN-CD successfully inhibited viral replication when administered at 8 h post-infection (hpi) on the apical surface of the HAE, IFN λ1 reduced viral spread only when administered at 24 h before infection (hbi), and on the basal side of the tissue (Supplementary Fig. 1a–c). Even though IFN λ pretreatment is not an ideal clinical option, our data are in line with already published in vitro and ex vivo studies, confirming the effectiveness of IFN λ only in pretreatment and on the basal side of polarized epithelial tissues[43,44]. The underlying reasons for that are the kinetic and the mechanism of action of IFN λ. Unlike 6'SLN-CD, which directly targets the virus and inactivates it within minutes[36], the antiviral state induced by IFN λ relies on the activation of a gene-expression program which takes several hours to be effective. Of note, in mouse models of infection, IFN λ prevents IV spread when administered via the intranasal route in therapeutic use, i.e., once the clinical symptoms of the disease are manifested, which would correspond to an administration at the apical side in our settings[26,30]. Of note, in vivo, IFN λ is sensed by the *trans*-epithelial dendritic cells of the respiratory mucosa, which strongly amplify its signal[45].

Based on these observations and to achieve the maximum combinatorial antiviral effect, we administered 6'SLN-CD and IFN λ1 according to the following protocol: HAE were first treated on their basal side with IFN λ1, starting at 24 hbi, while 6'SLN-CD was administered at 8 hpi on the apical side of the tissue. Both IFN λ1 and 6'SLN-CD were then co-administered daily up to 72 hpi (Fig. 1a). The quantification of viral replication at both 48 and 72 hpi, by measuring the viral particles released from the apical surface of the tissues, revealed that when administered in combination IFN λ1 and 6'SLN-CD

were more effective than when administered individually. The synergistic effect was evident at 48 hpi (≥1 log reduction for both individual treatments vs >2 log reduction for the combined one) and it persisted at 72 hpi, when the antiviral effect of the individual treatments was lost (Fig. 1b and Supplementary Fig. 1d). Of note, IFN λ1 and 6'SLN-CD are nontoxic nor as individual[32,36], nor as combined treatments (Supplementary Fig. 2). These data indicate that IFN λ1 treatment potentiates the antiviral action of 6'SLN-CD.

**scRNA-seq analysis reveals that the proportions of HAE basal, secretory and ciliated cells are affected neither by IAV infection nor by the antiviral treatments.** In order to investigate at the molecular level, the mechanism of action of IFN λ1 + 6'SLN-CD and its effects on HAE, we performed scRNA-seq analysis on both noninfected and infected tissues, administered or not with the individual or with the combined treatments. When conducting transcriptomic studies, it is essential to reach a fair compromise between viral and host gene expression. Viral replication occurs at the expense of the host transcription machinery, resulting eventually in a complete host shutoff[46]. Preventing the expression of cellular proteins at multiple steps is also a strategy adopted by the virus to counteract the antiviral response[47]. We selected the time of 48 hpi as the most suitable to perform scRNA-seq in our acute infection model, as it provides a wide window of analysis of both viral and host genes. At 48 hpi multiple cycles of infections already occurred and viral replication is in the exponential phase (Fig. 1b and[36]) resulting, however, in a still low cytopathic effect[11] and in ~10% infected cells, measured based on the expression of IAV nucleoprotein (NP) (Supplementary Fig. 3). Moreover, at this time point the effect of the combined treatment is significantly stronger than the individual ones that are however still efficient, allowing to compare therapeutic approaches with each other (Fig. 1b). When correlating, within the same HAE model, the number of IAV RNA copies measured from the apically released virus with the number of infected cells measured by FACS, we observed that the majority of the virus was produced by a small percentage of infected cells (Supplementary Fig. 3 and Fig. 1b). This finding is in line with previously published reports showing that between cells, there is a high level of variability in the outcome of IAV infection, resulting from multiple sources of heterogeneity, such as the number of viral transcripts per cell, the antiviral response and the timing of the infection[48,49].

sc-RNAseq relies on tissue dissociation, which can dramatically impact cell viability in epithelial tissues, as their survival is highly dependent on physical connections and communication between cells[50,51]. We established a dissociation protocol that allows us to retrieve every cell type of the HAE (secretory, basal, and ciliated cells, Supplementary Fig. 4) without compromising cell viability, thus preserving the quality of the mRNA within individual cells.

To perform scRNA-seq analysis, HAE were infected with IAV and treated or not with 6'SLN-CD, IFN λ1, or with IFN λ1 + 6'SLN-CD. To assess the perturbations induced by the formulation in the absence of a virus, an uninfected control (mock) untreated and one treated with IFN λ1 + 6'SLN-CD were included. Cells were partitioned for cDNA synthesis and barcoded using the Chromium controller system (10x Genomics), followed by library preparation and sequencing (Illumina). Sample demultiplexing, barcode processing, and gene counting were performed using the Cell Ranger analysis software[52].

The upper respiratory epithelium comprises several specialized cell types that likely respond to IAV infection in distinct ways[53]. Using Seurat analytical pipeline, we performed an unsupervised graph-based clustering[54] on the Cell Ranger integrated dataset, comprising all the tested experimental conditions (Fig. 2a–c). To match the identified clusters with the cell types found in the respiratory epithelium, we used both cluster-specific and canonical marker genes[55] (Fig. 2b, d and Supplementary Fig. 5a, b).

In all analyzed HAE we identified five distinct clusters. Three of them corresponded to mature basal (*TP63+/ITGA6+/KRT5* high/*KRT17* high), ciliated (*FOXJ1+/PIFO+/TPPP3+*), and a mixed population of secretory cells, including both goblet-mucous (*MUC5AC+*) and club cells (*SCGB1A1+/SCGB3A1+*) (Fig. 2a–d). One cluster was made of a population of basal cells uniquely defined by high levels of *LY6D*, a marker of cellular plasticity and differentiation[56] (Fig. 2a–d). Like in vivo, also ex vivo HAE basal cells have both self-renewing and multipotent properties[57,58]. The current consensus is that in steady-state conditions, basal cells differentiate first into secretory cells that in turn give rise to ciliated cells[59]. However, after injury, ciliated cells can be directly generated by basal cells[59,60]. *LY6D* high basal cells were characterized by the co-expression of both basal and secretory hallmark genes, such as *KRT5*, *KRT17*, and *BPIFB1*, *SPRR3*, *AGR2*, respectively (Fig. 2d). This cluster was hence identified as constituted by basal cells differentiating into secretory cells (BdiS). The fifth and last cluster, consisting of 843 cells (6.6% of the total selected cells), did not display a unique gene signature compared to the others, but co-expressed basal, secretory and ciliated hallmark genes (Fig. 2d and Supplementary Fig. 5b). It was also marked by an increased number of gene counts in comparison to the other clusters (Fig. 2e). We, therefore, concluded that this cluster likely resulted from doublets and excluded it from further analysis. These observed cell types and their proportions (Fig. 2f) are consistent with previous scRNA-seq studies and indicate that our ex vivo model recapitulates the respiratory epithelium in vivo[61].

We next determined the relative abundances of ciliated, secretory, basal, and BdiS across different conditions (Fig. 2f). We found that neither the infection alone nor the treatments in the presence or absence of the infection, induced substantial changes (>2 folds) in the relative proportions of these main epithelial cell types (Fig. 2f). IAV causes a strong cytopathic effect which results in a massive loss of ciliated cells and important alterations of the tissue structure. However, in our HAE infection model, this phenomenon occurs only at 120 hpi and it is therefore not evident at 48 hpi[11], which explains our results.

**The antiviral treatments affect IAV replication to different extents across different HAE cell types.** We next asked how viral transcripts would distribute across cell-type clusters, in each experimental condition. Global analysis of both host and viral transcriptomes in all 11,935 cells revealed that at 48 hpi and in the absence of treatments, IAV transcripts were detected in all cell types and were more abundant in ciliated cells, followed by secretory cells, BdiS and lastly, by basal cells (Fig. 3a–c). We classified the cells into four categories based on the frequency of viral transcripts (VT), i.e., the percentage of VT per cell: Z (cells bearing no VT), N (cells bearing a minimum non-zero frequency of VT, resulting from background noise), L (cells bearing VT at low frequency), M (cells bearing medium levels of VT), and H (cells bearing high levels of VT). Basal cells are located in the lower part of the epithelium, do not reach the apical side, and are therefore physically protected from the virus in the first stages of the infection when the ciliated cell layer is preserved[37]. Secretory cells have been shown to be the immediate target of IAV[11,62], while ciliated cells become preferentially infected at later stages of infection[63,64]. Nonetheless, we asked whether the different numbers of viral transcripts between secretory and ciliated cells relied also on the expression levels of host factors involved in IV infection. We measured in steady-state conditions the average mean expression of twelve cellular genes promoting multiple steps of IV replication[65], in secretory (including BdiS) vs ciliated cells (Supplementary Fig. 6). We found that secretory cells express higher levels of genes involved in IV RNA replication, such as CD151[66] and HMGB1[67], or in viral maturation and release like TMPRSS4[67] and Rack1[68]. While ciliated cells express higher levels of CLTA[69] and EPS8[70], necessary for viral endocytosis and uncoating (Supplementary Fig. 6). These data may provide further insights to understand the higher susceptibility to IAV infection of ciliated over secretory cells.

Compared to the mock steady state, in the 6'SLN-CD alone condition, all main epithelial cell types displayed a decreased number of viral transcripts. Even so, this reduction was more pronounced in secretory (~13.5 and 4.8-fold reduction in H and M cells, respectively) and in BdiS cells (~3.6 and 9-fold reduction in H and M cells, respectively), rather than in ciliated cells (~3.6 and 1.1-fold reduction in H and M cells, respectively) (Fig. 3a, b). Similarly, the treatment with IFN λ1 alone caused a greater reduction of viral replication in the non-ciliated compartment compared to the ciliated one (Fig. 3a, b). Of note, in all analyzed cell types, the inhibitory effect of IFN λ1 was stronger than that of 6'SLN-CD. Almost no viral reads were detected in the 6'SLN-CD + IFN λ1 condition, independently of the epithelial cell types (Fig. 3a, b). Accordingly, in the presence of both treatments, the number of infected cells measured by FACS accounted for less than 1% of the total epithelium (Supplementary Fig. 3). These results further confirmed the synergistic action of IFN λ1 and 6'SLN-CD and shed light on the cell-type-specific effects of the treatments. Interestingly, in the presence of 6'SLN-CD alone viral replication was hindered preferentially in secretory rather than in ciliated cells. This difference was probably determined by both IAV receptor specificity and the higher susceptibility to the infection of ciliated cells, which explains the stronger reduction of viral replication observed in secretory cells. As the effect of the IFN λ1 alone shows a similar trend across ciliated and secretory cells, the same explanation applies. We asked whether secretory cells mounted a stronger immune response compared to ciliated cells and measured, in both cell types, the average mean expression of several key ISGs, *OAS*, *MX1*, *MX2*, *IFIT1*, *IFIT2*, *ISG15*, and *ISG20*[20], across different experimental conditions.

Lastly, we investigated the expression levels of IAV mRNA segments, and we observed the following viral mRNA segment

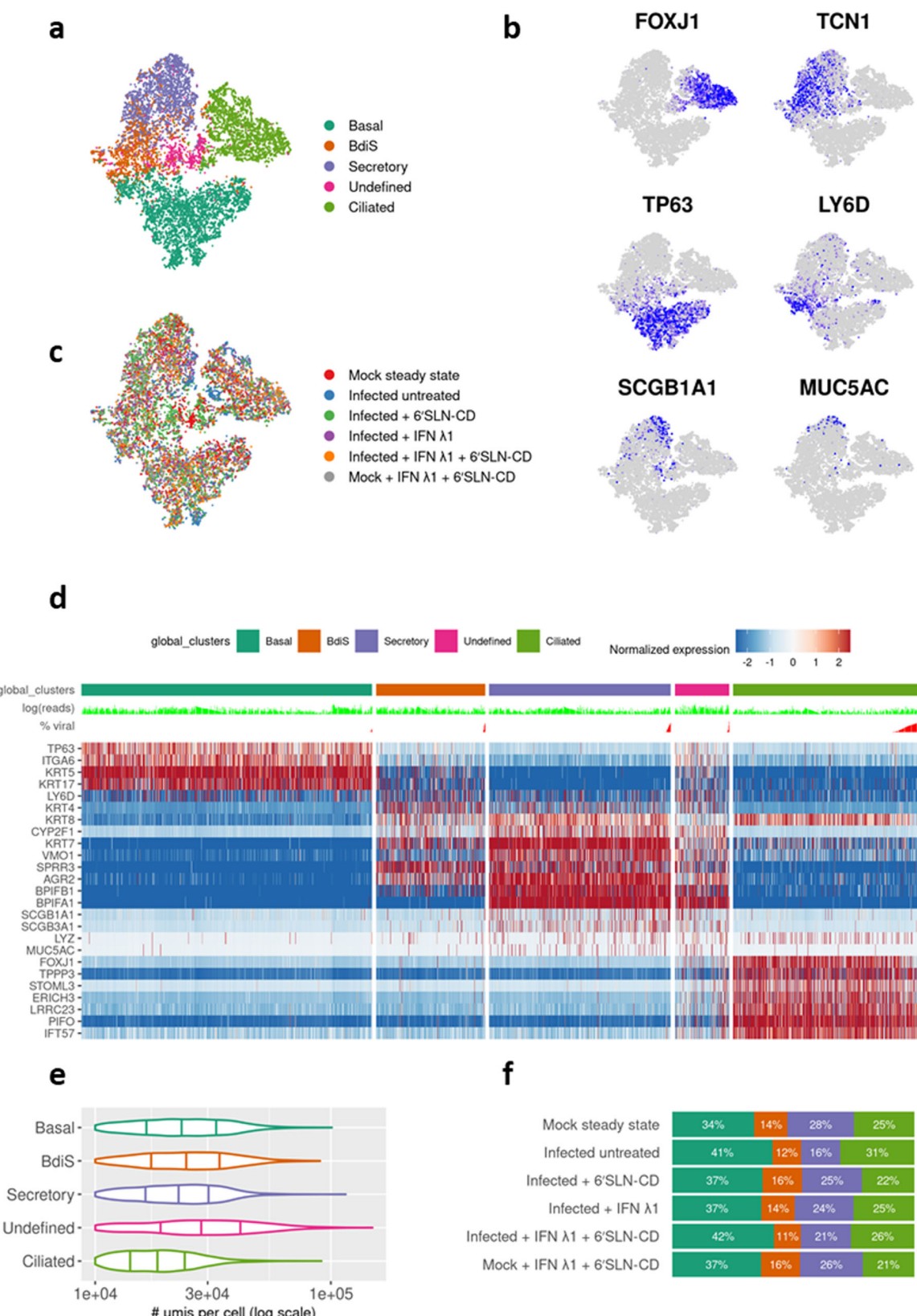

ratio: NEP > M2 > HA ~ NP > NA > M1 ~ NS1 > PA ~ PB1 ~ P2 (Supplementary Fig. 7). The fractions of individual viral genes did not change across the treatments (Supplementary Fig. 7). The spliced transcripts (M2 and NEP) had higher expression level compared to the unspliced transcripts (M1 and NS1). This finding is in line with previous reports showing that the expression of both M2 and NEP is more biased toward the later stages of viral replication, such as 48 hpi[64,71].

**Fig. 2 Analysis of HAE cell diversity. a** t-distributed stochastic neighbor embedding (t-SNE) visualization of the major cell types composing human HAE. Individual cell types were annotated using a combination of graph-based clustering results from Seurat and expression analysis of several canonical cell-type-specific markers. The t-SNE plots shown in panels **a–c** are presented in the same spatial orientation (i.e., the location of cells expressing the canonical markers in panel **b** corresponds to the location of the specific cell types in panel **a**. **b** t-SNE plots illustrating in blue the expression patterns of some of the canonical markers used to annotate the three main airway epithelial cell types: *FOXJ1* for ciliated cells, *TCN1* for all secretory cells, *TP63* for basal cells, *LY6D* for differentiating cells, *SCGB1A1* for club cells and MUC5AC for goblet cells; scale bars are Log2. **c** t-SNE visualization of the scRNA-seq data for all single cells in the following conditions: mock steady sate, infected untreated (infected with A/Switzerland/3076/2016 H1N1 strain), infected + 6'SLN-CD, infected + IFN λ1, infected + IFN λ1 + 6'SLN-CD, and mock + IFN λ1 + 6'SLN-CD. **d** Heatmap representing the gene-expression profiles of 12,778 single cells from human HAE grouped into five clusters. The percentages of viral reads across cells are shown in red above the heatmap, while the number of total UMI counts is shown in light green. In each cluster, infected cells are ordered by increasing number of viral reads. Expression values are Pearson residuals from SCTransform binomial regression model [70] fitted to UMI counts (see Methods). The cells were clustered solely on the expression of the shown hallmark genes; HAE human airway epithelia, BdiS basal differentiating into secretory cells. **e** Violin plots showing the distributions of per-cell UMI (unique molecular identifiers) counts in HAE cell clusters. **f** Bar graph showing the relative percentage of each main epithelial cell type described above in each experimental condition described in **c**.

**scRNA-seq analysis reveals cell-type-specific responses to the infection and to the treatments within each HAE cell cluster.** We then sighted to further investigate the heterogeneous cell responses to IAV infection and to the treatments within each epithelial cell cluster. Individual clustering[54] was performed by analyzing each main HAE cell type independently of the others and led to the identification of several subpopulations, or subclusters.

Basal cells were distributed across six subclusters annotated as follows: (b1) steady-state basal cells; (b2) and (b3) *LY6D+* differentiating cells[55], with b3 displaying a more pronounced expression of *KRT14* and *KRT16*, markers of tissue repair and regeneration[72,73]; (b4) highly proliferating cells, based on strong expression levels of genes involved in cell cycle progression such as *MIKI67*, *CDK1*, and *BIRC5*; (b5) proliferating cells with lower levels of cell cycle progression genes, compared to b4, but with high levels of *KRT14* high and lastly (b6) inflamed cells, based on high expression of *CXCL10*, *CXCL11* and several others ISGs (Fig. 4a and Supplementary Fig. 8a). The latter subcluster was the less represented in the mock steady-state control, while it became the most abundant in the infected-untreated condition (16-fold increase), with an inflammation signature stronger than that induced in the other subclusters (Fig. 4a and Supplementary Fig. 8a). IAV induced the expression of pro-inflammatory cytokines across all basal subpopulations. 6'SLN-CD and IFN λ1, administered alone or in combination, counteracted this effect (Supplementary Fig. 8a). Similarly, the b6 inflamed cluster was decreased by 3-fold by the individual treatments and by 9-fold by the combined formulation. In turn, the levels of differentiating basal clusters (b2 and b3), which were decreased by the infection (2-fold and 3-fold decrease for b2 and b3, respectively), were also restored by the antivirals. In line with previous reports[74], IAV infection also reduced the b4 highly proliferating subcluster (2.8-fold decrease), which was not recovered by the individual, nor by the combined treatments (Fig. 4a). This may also result from the inflammatory response triggered by IFN λ1, as b4 is less abundant also in the mock double treated, compared to the mock steady state (2-fold decrease). On the other hand, the b5 low proliferating cells cluster did not undergo strong changes across the tested experimental conditions.

Of note, as they are not a direct target of IAV (Fig. 3a and Supplementary Fig. 8a), basal cells mainly contributed to the immune reaction against the virus as bystander cells[75]. Thus, all the changes induced by IAV in this epithelial compartment were largely independent of viral replication.

BdiS cells, characterized by the expression of *LY6D*[55], comprised six subclusters, sharing an overall common transcriptional profile, with the exception of few genes: (bd1) steady-state cells; (bd2) SERPINB3 high cells; (bd3) CXCL1 high cells,

displaying a strong expression of CXCL1, but not increased levels of other inflammatory cytokines, compared to bd1; (bd4) and (bd5), characterized by unique high expression of the intermediate filaments genes KRT24 and KRT23, respectively, and, lastly bd6) inflamed BdiS cells, based on the expression of several ISGs (Supplementary Fig. 8b). Similarly to basal cells, IAV infection caused in BdiS an increase of the inflamed cluster (14-fold increase), at the expenses of the others. The antiviral treatments strongly reduced the fraction of bd6 (Fig. 4b). Of note, while mostly acting as bystander cells, BdiS cells supported viral replication more than basal cells (Fig. 3), suggesting that the transition into secretory cells is accompanied by an increased permissiveness to the infection.

Secretory cells were classified in three subclusters: (s1) steady-state secretory cells comprising a mixed population (defined as mixed because the gene-expression profiles did not allow unambiguous classification) of mainly club cells and fewer goblet cells, displaying a high expression of *SCGB1A1*, *SCGB3A1*, *MUC5AC*, *RARRES1,* and *LCN2*; (s2) *SERPINB3/PI3* high cells, and (s3) inflamed cells, based on higher expression levels of ISGs (Fig. 4c and Supplementary Fig. 8c). IAV infection triggered the expression of pro-inflammatory cytokines in all secretory subpopulations; however, this effect was stronger in the s3 subcluster, whose fraction was increased by 10.8-fold, at the expenses of the others (Fig. 4c and Supplementary Fig. 8c). Secretory cells represent the second target of IAV after ciliated cells. We did not observe an additive effect of the treatments in secretory cells: compared to the infected-untreated condition, IFN λ1 alone decreased the fraction of s3 inflamed cells by only 1.89-fold, while 6'SLN-CD alone restored the secretory subclusters composition nearly as effectively as the combined treatments. This is probably due to the fact that, similarly to basal and BdiS cells, most of the changes occurring in secretory cells after IAV infection were largely independent of viral replication.

Within the ciliated compartment, we identified four subclusters: (c1) steady-state cells with high expression of ciliated hallmark genes *FOXJ1*, *TPPP3*, and *ERICH3*; (c2) immature cells, based on lower levels of the ciliated hallmark genes, and on higher expression of *RAB11FIP1*, which is involved in primary ciliogenesis[76]; (c3) *IFN −* inflamed cells; and (c4) *IFN +* inflamed cells, both characterized by high expression of inflammatory genes, such as *ISG20* and *GBP1*, but differing from each other based on the expression of *IFN λ* and *IFN β1* (Fig. 4d and Supplementary Fig. 8d).

Ciliated cells are highly permissive to IAV infection[63]. Analyzing the distribution of viral transcripts, we found that viral replication occurred across all ciliated subclusters (Supplementary Fig. 8d). However, c4 displayed the highest levels of IAV segments, resulting in 100% of infected cells (Supplementary

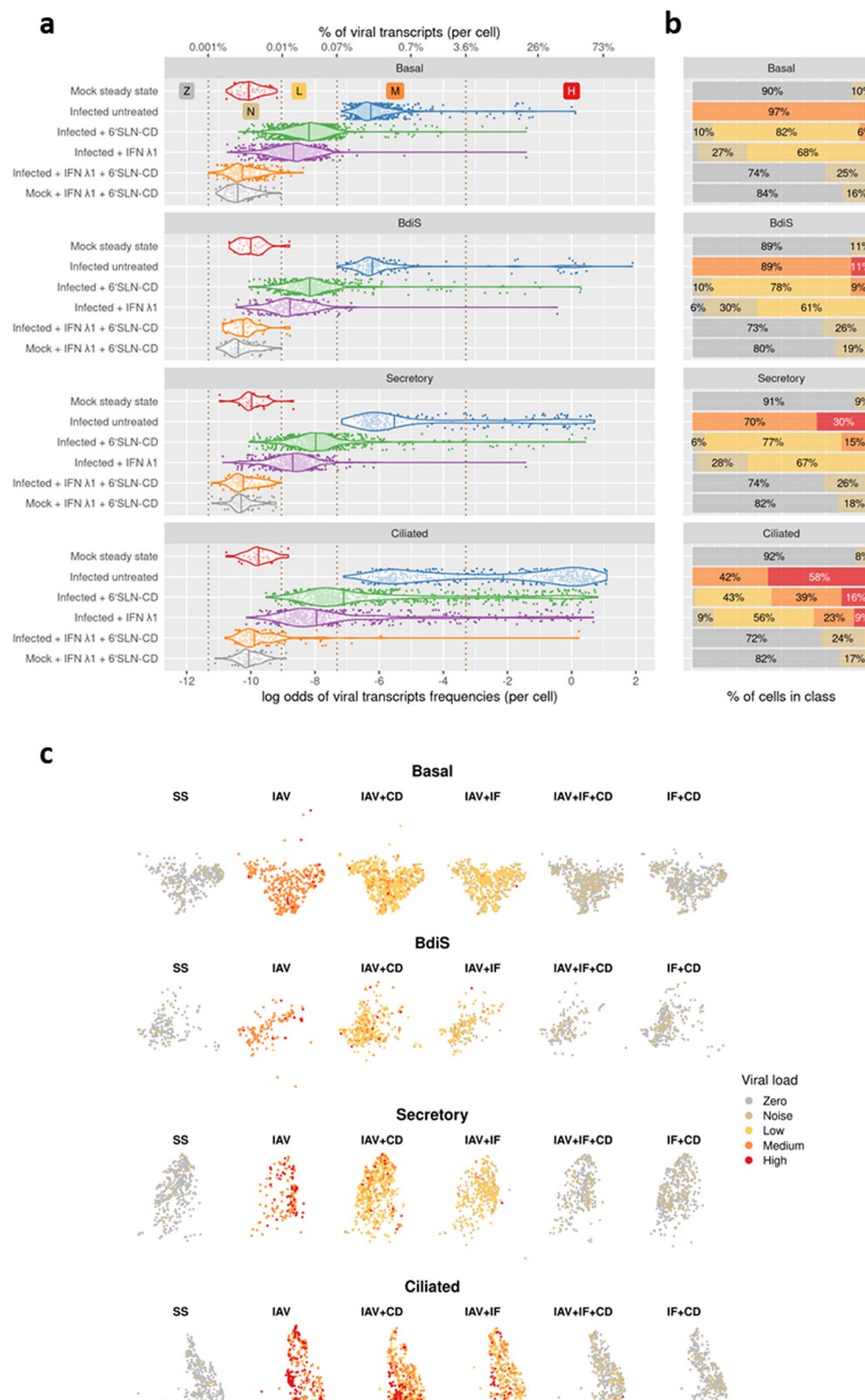

**Fig. 3 Within-cell viral load across cell types and conditions (log odds scale). a** Violin plots of within-cell proportions of viral transcripts (VT) on log odds scale (by cell type and condition). Vertical dashed lines define different viral load classes: Z = zero VT; N = background noise of VT; L = low VT; M = medium VT; H = high VT (see Methods). **b** Fractions of cells in viral load classes within cell-type and condition groups. The thresholds defining the viral load classes as per (see methods) are shown as vertical dashed lines in panel **a**. **c** t-distributed stochastic neighbor embedding (t-SNE) visualization of fractions of cells in viral load categories, within cell-type and condition groups. Conditions as specified in Fig. 2c.

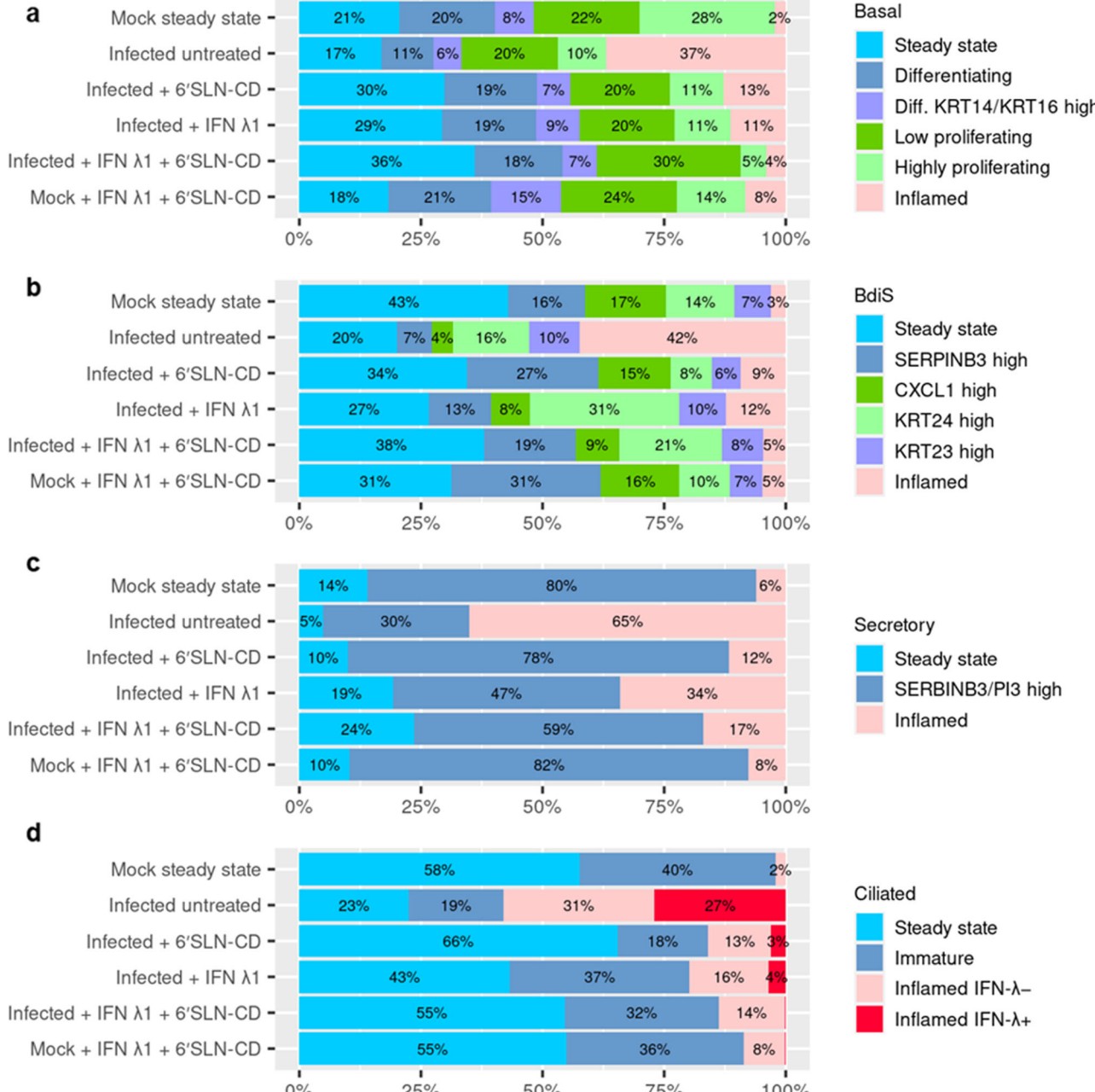

**Fig. 4 Analysis of HAE cell types heterogeneity.** Stacked bar graph showing the relative percentages of HAE basal (**a**), **b** BdiS (basal differentiating into secretory), **c** secretory, and **d** ciliated subclusters across experimental conditions. Conditions as specified in Fig. 2c. HAE = human airway epithelia.

Fig. 8d). The viral load correlated with the entity of the inflammatory response, as only c4 expressed *IFN λ* and *IFN β1* genes, as well as high levels of *NEDD9*, which is associated with IAV-induced antiviral response[77]. Moreover, compared to all other clusters, c4 exhibited high levels of the proapoptotic factor *BBC3*, and lower or null levels of ciliated hallmark genes, probably as a result of the massive viral genes expression, hijacking the host transcriptional machinery[46] (Supplementary Fig. 8d). The changes in the proportions of ciliated cells subclusters across experimental conditions reflected the efficacy of the treatments. Compared to the mock steady-state control, IAV infection resulted in a relative decrease of both the steady-state and immature cells (2.6-fold reduction of c1 and 2.1-fold reduction of c2, respectively), whose levels were restored by both individual and combined treatments (Fig. 4d). In line with that,

6'SLN-CD and IFN λ 1 counteracted the increase in the c3 inflamed IFN-subcluster (Fig. 4d). The C4 inflamed IFN + subpopulation followed a similar trend, but completely disappeared in the 6'SLN-CD + IFN λ 1 condition, further proving the combinatorial effect of the two compounds. Interestingly in the mock-treated condition, the inflamed IFN − cluster was increased compared to the mock steady state but not the inflamed INF+, indicating that the latter represents a virus-specific signature at 48 hpi (Supplementary Fig. 8d).

Our findings show that IAV infection alters the subclusters composition in epithelial cell type by inducing the appearance of inflamed populations. As the inflammatory response tightly correlates with the viral load, the ability of the antiviral treatments to restore the tissue composition to the steady-state level is stronger in infected rather than in bystander epithelial cell types.

## Discussion

Influenza can be a dreadful disease, with a strong socio-economic impact worldwide. Current antiviral strategies are only efficient when administered within a short time after the onset of the symptoms and are challenged by the genomic instability of the virus. We are in need of additional antiviral therapies targeting the respiratory immune defense to improve viral clearance, reduce the risk of bacterial super-infection, and attenuate tissue injury. A treatment that would prevent viral entry and, at the same time, boost the host antiviral response without causing immunopathology would thus represent an ideal tool to prevent or treat influenza infection. With this in mind, we assessed the antiviral potential of co-administering human IFN λ1 with 6'SLN-CD against H1N1 IAV in ex vivo HAE. The IFN λ1 + 6'SLN-CD formulation is nontoxic and more effective in reducing viral replication, compared to the individual treatments. IFN λ1 has already been used in clinical trials against viral infections, while 6'SLN-CD is well tolerated in vivo and effectively constrains the spread of IV infection when administered topically[36]. Overall, our data support a prospective therapeutic application of IFN λ1 + 6'SLN-CD.

We next sought to investigate the investigated transcriptomic impact of this formulation by scRNA-seq in HAE. Transcriptomic analysis unraveled the heterogeneous composition of each main epithelial cell type, which is an assortment of subclusters with unique gene-expression programs underlying different cell states. Besides terminally differentiated cells, we also identified BdiS cells. This subpopulation, roughly equally represented across experimental conditions, derived from the continuous differentiation process occurring in the respiratory epithelium. We did not find basal cells differentiating directly into ciliated cells, a process triggered by tissue injury[58,78,79], because, in our settings, the cytopathic effect induced by IAV at 48 hpi is not strong enough to alter the architecture of the tissues[11]. We also did not observe secretory cells differentiating into ciliated cells[54–56], probably due to the limitations in sequencing depth and the fact that we did not perform a lineage study[57], which would be beyond the scope of this work. When we measured the distribution of viral reads across the main cell types, we found that IAV preferentially infected epithelial cells in the following order: ciliated, secretory, and basal cells. Accordingly, ciliated cells mounted a stronger inflammatory response compared to secretory cells, which, in turn, expressed ISGs and innate cytokines at higher levels than basal cells. Interestingly, only within the ciliated compartment, IAV induced the appearance of highly inflamed cells characterized by a distinctive high expression of IFN type I and type III genes. Basal and BdiS cells, which were the most diverse due to their multipotent potential, displayed extremely low levels of viral transcripts and participated in the tissue immune response as bystander cells. Of note, the infection of basal cells would be highly detrimental to the host as these cells are absolutely necessary to maintain the barrier of the respiratory epithelium by regenerating secretory and ciliated cells targeted by IV[79]. We observed that the proportions of the main HAE cell-type clusters did not change upon infection and/or treatments which is, as explained above, due to the poor cytopathic effect induced by IAV at 48 hpi[11]. On the other hand, the subcluster composition of each cell type underwent strong modifications in response to the infection, mostly resulting from the appearance of inflamed cells. These changes were induced in ciliated and to some extent in secretory cells, by a direct cell response to viral replication, while in basal and BdiS cells by the response to the paracrine signaling from infected cells.

The individual treatments reduced the percentage of viral transcripts to different extents across epithelial cell types: both 6'SLN-CD and IFN λ1 alone caused a reduction of viral reads more pronounced in secretory rather than in ciliated cells. This finding was unexpected for 6'SLN-CD, which is designed to exclusively target extracellular viral particles and was evenly distributed throughout the apical side of the HAE. We reasoned that the effect of the 6'SLN-CD in reducing viral replication depends on the susceptibility of the cells to the infection, which is dictated by both the distribution of α 2,6-linked Sia and the expression of host factors necessary for viral entry. As IAV infects ciliated cells more easily than secretory cells, the number of viral transcripts is lower in the latter cell type, and in turn, its reduction in response to 6'SLN-CD is more pronounced, compared to that observed in ciliated cells. Of note, the increase of inflamed cells observed in secretory cells upon the infection is higher than that observed in ciliated cells (Fig. 4c, d), even though the latter are the actual source of virus. This finding is in line with the sequential infection of secretory and then ciliated cells. Probably IAV first massively infects secretory cells, that mounts a strong inflammatory response without being able, however, to support viral replication, and in a second step prefers ciliated cells.

Viral transcript distribution analysis also indicated a synergistic effect between 6'SLN-CD and IFN λ1 in each main cell type. However, the capacity of the combined treatments to revert IAV-induced perturbations in subclusters composition was greater than the individual ones only in ciliated cells, where the inflammation was a direct consequence of viral replication and in basal cells, but only limited to the inflamed subcluster. In line with that, in the basal compartment, where changes in cell composition were independent of IAV infection, not the individual nor the combined treatments succeeded in restoring the number of highly proliferating cells.

Lastly, in the absence of infection, the combined treatment did not alter the ratios between the main basal, ciliated, and secretory cells clusters, but changed the subclusters abundances within each of them, resulting in an increase in inflammatory cells, which was likely induced by IFN λ1.

Different macromolecule-based approaches are currently available for the treatment of viral infections. However, a deep knowledge of the impact on the host cells is needed to increase the effectiveness of these therapies, minimize the side effects and reduce the toxicity. Our study, proposing scRNA-seq to assess the effects of a combined therapy against IV, is in line with this need and, to the best of our knowledge, is the first to present such an analytical approach. We suggest that the ability of an antiviral treatment to restore epithelial cell subclusters composition upon infection strongly correlates with that reducing the inflammatory response. This is an important parameter to dissect the effects of the drug on the host cells, beyond its capacity to impair viral replication. Our work is also the first in addressing at the molecular level the anti-IAV effects of IFN λ in HAE. Additional investigations in more relevant in vivo models of infection, such as mice or ferrets, will be necessary to further assess the efficacy of IFN λ1 + 6'SLN-CD formulation, with a more realistic administration protocol where both compounds are administered in post-treatment, and its genetic barrier to antiviral resistance. Also, in light of the ongoing differentiation process occurring in the respiratory epithelium, scRNA-seq velocity analysis[80] could allow for investigation of the trajectories of both basal and secretory cells differentiation, as well as how such trajectories would be perturbed by the infection and the treatments.

## Methods

**Human airway epithelia (MucilAir™).** The human airway epithelia used in this study were reconstituted from freshly isolated primary human nasal polyp epithelium collected either from 14 different donors, upon surgical nasal polypectomy, or from individual donors, as previously described[11]. The patients presented with nasal polyps but were otherwise healthy, with no atopy, asthma, allergy, or other

known comorbidities. All experimental procedures were explained in full, and all subjects provided written informed consent. The study was conducted according to the Declaration of Helsinki on biomedical research (Hong Kong amendment, 1989), and the research protocol was approved by the local ethics committee[11]. The tissues were maintained at the air-liquid interface according to the manufacturer's instructions[11].

**Viral stocks and compound.** Influenza H1N1 A/Switzerland/3076/16 clinical specimen was isolated from the nasopharyngeal swab of an anonymized patient, provided from the Geneva University Hospital. The sample was screened by one-step real-time quantitative PCR (qPCR)[81]. Influenza A virus was subtyped by sequencing the NA gene as previously described[82]. To prepare viral stocks, 100 μl of the clinical sample was inoculated at the apical surface of several HAE, for 4 h at 33 °C. After the infection, the apical side of the tissues was washed five times with PBS. In order to measure the daily viral production, apical samples were collected every 24 h, by applying 200 μl of medium for 20' at 33 °C. The viral load of each time point was then measured by RT-qPCR, and the four apical washes with the highest titer were pooled and re-quantified. Aliquots were stored at −80 °C.

Human recombinant IFN λ1 protein was obtained from R&D Systems, Inc. (Abingdon, United Kingdom). 6'SLN-CD was synthesized as described previously[36].

**HAE, viral infections, and treatments.** HAE were infected apically with H1N1 A/Switzerland/3076/16 strain (1e3 or 1e4 RNA copies/tissue), in a final volume of 100 μl as described above[83]. Infected tissues were treated with 6'SLN-CD alone, IFN λ1 alone, or with 6'SLN-CD plus IFN λ1. 6'SLN-CD dissolved in PBS was transferred on the apical surface of the tissues (60 μg/tissue, in a volume of 30 ul), starting from 8 hpi. After each apical wash, performed for daily viral load quantification as described above, 6'SLN-CD was re-added on the apical side of the tissues. IFN λ1 was added on the basal side of the inserts (5.5 ng/tissue in 550 μl) one day before infection and then added every day, each time replacing the entire basal medium with a fresh one. In parallel, upon each wash, the infected-untreated tissues were administered with 30 μl of PBS on the apical side, the same volume added to the tissues treated with 6'SLN-CD, while the basal medium was changed on a daily basis. The treatments were administered up to 72 hpi.

**Viral load quantification.** Viral RNA was extracted from Mucilair apical washes using EZNA viral extraction kit (Omega Biotek) and quantified by using RT-qPCR with the QuantiTect kit (#204443; Qiagen, Hilden, Germany) in a StepOne ABI Thermocycler, as previously described[11]. The primers used are described in Table 1. Viral RNA copies were quantified as follows: four ten-fold dilution series of in vitro transcripts of the influenza A/California/7/2009(H1N1) M gene were used as reference standard as previously described[11]. CT values were converted into RNA load using the slope-intercept form. In all experiments, the slope, efficiency, and R2 ranged between 0.96 and 0.99[38,84]. *P*-values were calculated relative to untreated controls using the two-way ANOVA with Prism 8.0 (GraphPad, San Diego, CA, USA).

In addition to RT-qPCR, viral replication in human airway epithelia was quantified by plaque assay. MDCK cells (a kind gift of Prof. Mirco Schmolke, University of Geneva) 500000 cells per well, were seeded in six-well plate 24 h in advance. Serial dilutions were added to cells in a final volume of 500 μl for 1 h at 37 °C. MDCK monolayers were then washed and overlaid with 0.8% agarose in a medium supplemented with TPCK trypsin 1 μg/ml. Two days after infection, cells were fixed with paraformaldehyde 4% and stained with crystal violet solution containing ethanol. Plaques were counted.

**Toxicity and viability assays.** Noninfected tissues were treated with 6'SLN-CD plus IFN λ1, in the same doses/volumes used for infected tissues (as described above). Accordingly, every day an apical wash was performed, and a new dose of 6'SLN-CD was added on the apical side of the tissues, while a fresh medium with IFN λ1 was added on the basal side. Similarly, the untreated control tissues were administered with 30 μl of PBS on their apical side, while the basal medium was replaced every day.

Lactate dehydrogenase (LDH) release in the apical medium was measured with the Cytotoxicity Detection Kit (Roche 04744926001) as described previously[36]. Percentages of cytotoxicity were calculated compared to the cytotoxicity control tissues, which were treated with 100 μl of PBS-5% Triton ™ X-100 (Sigma–Aldrich) on the apical side.

Cell metabolic activity was measured by adding MTT reagent (Promega), diluted in MucilAir medium (1 mg/ml), on the basal side of the tissues. After 3 h at 37 °C the tissues were transferred to a new plate and lysed with 1 ml of DMSO. Subsequently, the absorbance was read at 570 nm according to manufacturer instructions. Percentages of viability were calculated by comparing the absorbance to the untreated control tissues.

**HAE enzymatic dissociation and flow cytometry analysis.** HAE tissues were first washed both apically and basally in DPBS without calcium and magnesium (Thermo Fisher Scientific) for 10' at 37 °C. Then they were incubated with TrypLE (Thermo Fisher Scientific), both apically and basally, for 30' at 37 °C. During this time, the tissues were dissociated with a 1 ml pipette. Cells were harvested and washed with ice-cold MACS buffer (PBS without calcium and magnesium, EDTA pH 8, 2 mM BSA 0.5%).

For scRNA-seq, cells were stained with Hoechst 33342 (Thermo Fisher Scientific) and DRAQ7 (Biolegend) and analyzed with a MoFlo Astrios Cell Sorter (Beckman Coulter). Viable cells were defined as Hoechst+/DRAQ7-, doublets were excluded by gating for SSC-W vs SSC, and single cells were sorted.

To determine the percentages of infected cells, upon HAE dissociation, the cells were fixed/permeabilized using the Perm/Wash Buffer RUO (554723 BD Biosciences-US) and then stained with the primary antibody (mouse monoclonal anti-IVA Ab 1:100 dilution, Chemicon®) for 20' at 4 °C. After a wash with Perm/Wash Buffer RUO the secondary Ab (Alexa Fluor 488 Invitrogen™, 1:200) was added for 20' at 4 °C. After one wash with MACS buffer the percentages of IAV infected cells were determined with a MoFlo Astrios Cell Sorter (Beckman Coulter) and the uninfected gating control was defined using uninfected cells stained with both the primary and the secondary antibodies.

**Single-cell RNA-sequencing of HAE.** Upon HAE dissociation, viable cells were sorted as described above. Cells were then counted using Countess™ II FL Automated Cell Counter (Invitrogen) and diluted to equivalent concentrations with an intended capture of 5000 cells/sample, following the manufacturer's provided by 10x Genomics for the Chromium Single Cell platform. All subsequent steps through library preparation followed the manufacturer's protocol. Samples were sequenced on an Illumina HiSeq 4000 machine.

**Table 1 Primers used for influenza viral load quantification.**

| Target gene | Influenza A virus neuraminidase |
|---|---|
| H1N1 probe | TGCAGTCCTCGCTCACTG GGCACG |
| H1N1 forward | GACCRATCCTGTCACCTCTGAC |
| H1N1 reverse | AGGGCATTYTGGACAAAKCGTCTA |

**Table 2 *P*-values from tests of cell-type or subcluster proportions being the same across conditions.**

| Contrasted conditions | | Cell types | Subclusters | | | |
|---|---|---|---|---|---|---|
| Condition A | Condition B | | Basal | BdiS | Secr. | Cil. |
| Infected untreated | Mock steady state | <0.001 | <0.001 | <0.001 | <0.001 | <0.001 |
| Infected + 6'SLN-CD | Infected untreated | <0.001 | <0.001 | <0.001 | <0.001 | <0.001 |
| Infected + IFN λ1 | Infected untreated | <0.001 | <0.001 | <0.001 | <0.001 | <0.001 |
| Infected + IFN λ1 + 6'SLN-CD | Infected untreated | 0.002 | <0.001 | <0.001 | <0.001 | <0.001 |
| Infected + 6'SLN-CD | Infected + IFN λ1 | 0.012 | 0.605 | <0.001 | <0.001 | <0.001 |
| Infected + IFN λ1 + 6'SLN-CD | Infected + 6'SLN-CD | <0.001 | <0.001 | <0.001 | <0.001 | <0.001 |
| Infected + IFN λ1 + 6'SLN-CD | Infected + IFN λ1 | <0.001 | <0.001 | 0.002 | <0.001 | <0.001 |
| Infected + IFN λ1 + 6'SLN-CD | Mock steady state | <0.001 | <0.001 | 0.064 | <0.001 | <0.001 |
| Mock + IFN λ1 + 6'SLN-CD | Mock steady state | 0.011 | <0.001 | 0.001 | 0.150 | <0.001 |

The table shows p-values from a Chi-squared contingency test on cell-type (subcluster) counts across condition pairs. The null hypothesis is that the cell-type (subcluster) distributions are independent of the conditions. The p-values were obtained by sampling the Chi-squared statistic under the null hypothesis (one million times for each comparison).

**Table 3 Definition of viral load classes used in Fig. 3 and throughout section "The antiviral treatments affect IAV replication to different extents across different HAE cell types".**

| Classes | | Threshold definition | Value |
|---|---|---|---|
| **Below** | **Above** | | |
| Zero | Noise | Global minimum non-zero frequency of viral transcripts (FVT) | 0.0012% |
| Noise | Low | The 99th percentile of FVTs in uninfected cells not in Zero | 0.012% |
| Low | Medium | Minimum FVT across infected-untreated cells | 0.066% |
| Medium | High | The 75th percentile of FVTs across infected-untreated cells | 3.57% |

The table shows the definitions of the thresholds dividing the range of possible within-cell viral transcript frequencies into five adjacent intervals (viral load classes). The Undefined global cluster was excluded while computing the thresholds.

**Table 4 Average within-cell frequencies of IAV transcripts across cell types and conditions.**

| | Basal | BdiS | Secretory | Ciliated |
|---|---|---|---|---|
| Uninfected untreated | 0.0005% (0.0004–0.0007%) | 0.0005% (0.0003–0.0009%) | 0.0005% (0.0003–0.0007%) | 0.0005% (0.0003–0.0008%) |
| Infected untreated | 0.89% (0.76–1.04%) | 6.1% (4.3–8.7%) | 10% (8–13%) | 22% (19–25%) |
| Infected + 6'SLN-CD | 0.068% (0.062–0.075%) | 0.62% (0.49–0.79%) | 0.56% (0.46–0.67%) | 5.7% (4.7–6.8%) |
| Infected + IFN λ1 | 0.053% (0.046–0.062%) | 0.22% (0.15–0.30%) | 0.068% (0.057–0.083%) | 3.3% (2.5–4.2%) |
| Infected + IFN λ1 + 6'SLN-CD | 0.0010% (0.0009–0.0012%) | 0.0010% (0.0007–0.0014%) | 0.0010% (0.0008–0.0013%) | 0.16% (0.09–0.28%) |
| Mock + IFN λ1 + 6'SLN-CD | 0.0006% (0.0004–0.0007%) | 0.0008% (0.0005–0.0010%) | 0.0007% (0.0005–0.0009%) | 0.0009% (0.0007–0.0012%) |

The table shows the point estimates and the 99% confidence intervals from a Beta-Binomial regression model estimated (see Methods). The estimates from the Beta-Binomial model are naturally on log odds scale and were converted to probabilities for presentation in this table. Note that the difference of parameter estimates is not necessarily insignificant if the respective confidence intervals overlap—such an assessment is overly conservative.

**Computational analysis of scRNA-seq data**. Upon demultiplexing and performing the routine quality checks on the raw reads, we processed the data via Cell Ranger version 3.1.0 using the union of human and Influenza A genome as a reference (see References and Annotations) (https://support.10xgenomics.com/single-cell-gene-expression/software/overview/welcome). We extracted the UMI counts for the 10,000 most frequent cell barcodes in each sample, then screened the distributions of total UMI counts, percentages of mitochondrial and viral genes across these barcodes (within each sample), and, selected cell barcodes having more than 10,000 UMIs[85]. This selection procedure resulted in 13805 selected cells across all samples with (a) 100,222 raw reads per cell on average, (b) median UMI count per cell from 17,031 to 27,419 depending on the sample, (c) the median number of genes per cell from 3998 to 5223 depending on the sample. The percentage of reads mapped confidently to the genome (transcriptome) varied in the range from 85.3 to 87.3% (61.2–63.9%). We then excluded cells where more than 15% of UMI counts correspond to mitochondrial genes, which resulted in 12,778 captured cells for downstream analysis (~2129 cells per condition). Of note, since our partitioning input was 4000 cells per condition, the recovery rate was about 50%, which is in line with previously reported works[52].

The analysis of single-cell data was performed using Seurat version 3.2.3[86]. First, the raw UMI counts were transformed to normalized expression levels on a common scale using SCTransform method implemented in Seurat, which amounts to computing (Pearson) residuals in a regularized binomial regression model for UMI counts[87]. The normalization was performed jointly on all samples, and the genes expressed in less than 10 cells were excluded prior to normalization together with the viral genes.

The selected cells from all samples were first clustered on the normalized expressions of hallmark marker genes only, which resulted in five stable clusters. The clustering method implemented in Seurat we applied amounts to (1) constructing a $k$-nearest neighbor graph of all cells, (2) deriving an (approximate) shared nearest-neighbors graph, and (3) applying a modularity-based community detection to the latter graph[88]. Euclidian metric was used for the construction of $k$-NN graph.

The five identified global clusters lacked a clear signature and contained a sizeable proportion of cells with high total UMI counts compared to other clusters (Fig. 3). Given it included only a small percentage of cells, we excluded it from further analysis, hypothesizing that this cluster likely contains a large percentage of doublets. The remaining four clusters were clearly identified as basal cells, ciliated cells, secretory cells, and basal cells differentiating into secretory cells (Fig. 3 and Supplementary Fig. 6). The latter two clusters were merged for further analysis.

The cells in the identified global cell-type clusters were then analyzed in isolation from each other in order to identify cell-type-specific responses. For each cell type, we first found a tentative set of genes differentially expressed between conditions and then reclustered the cells based on their expressions across these genes. The same graph-based clustering was applied with cell distances derived from the first 10 principal components of the expression matrix. Condition-differential genes were found as a union of genes overexpressed in any condition versus the rest as assessed by the Mann–Whitney–Wilcoxon test with the nominal $p$-value of 0.01. The obtained differential genes were ordered using a hierarchical clustering algorithm and manually curated before producing the cell-type-specific heatmaps shown in Supplementary Fig. 8. All our scRNA-seq data are deposited to GEO (GSM5740432).

**Analysis of cluster and subcluster compositions across conditions**. to assess whether the proportions of identified HAE cell-type clusters (proportions of identified subclusters within the cell types) were materially different across conditions, we used a Chi-squared contingency in the following comparisons: (1) infected untreated vs mock steady state, (2) each infected treated condition vs infected untreated condition, (3) infected IFN λ1 vs infected 6'SLN-CD, (4) infected combined treatment vs each of the individual treatments, (5) infected combined treatment vs mock steady state, and, finally (6) mock combined treatment versus mock steady state. The $p$-values were computed by sampling the Chi-squared statistic under the null hypothesis i.e., the joint distribution being a (scaled) product of the marginal distributions (one million samples were generated for each comparison).

The results of the tests are presented in Table 2. The nulls that the cell-type distributions or subcluster distributions are independent of conditions were, generally, strongly rejected (with a few exceptions). For cell-type composition tests, the differences between the two individual treatments (in infected cells) were not significant ($p$-value of 0.012), and the same was observed for the differences between the mock-treated condition and the steady state ($p$-value of 0.011). For the tests of subcluster compositions, the null of no differences between the mock-treated condition and the steady state was rejected for three out of four cell types suggesting the test might be overly conservative. Indeed, the subclusters were identified by clustering on the condition-differential genes identified in the cell types (in the very same dataset).

**Analysis of host factors**. To test for the differential expression of host factors between secretory and ciliated cells in steady-state conditions, we first filed a list of

**Table 5 P-values from tests of differences in viral load across cell types in the same condition.**

| Condition | BdiS vs basal | | Secretory vs BdiS | | Ciliated vs secretory | |
|---|---|---|---|---|---|---|
| | BB | Chi2 | BB | Chi2 | BB | Chi2 |
| Uninfected untreated | 0.934 | 0.291 | 0.601 | >0.999 | 0.667 | 0.605 |
| Infected untreated | <0.001 | <0.001 | 0.002 | <0.001 | <0.001 | <0.001 |
| Infected + 6'SLN-CD | <0.001 | <0.001 | 0.309 | 0.005 | <0.001 | <0.001 |
| Infected + IFN λ1 | <0.001 | 0.117 | <0.001 | 0.503 | <0.001 | <0.001 |
| Infected + IFN λ1 + 6'SLN-CD | 0.935 | 0.615 | 0.937 | 0.123 | <0.001 | <0.001 |
| Mock + IFN λ1 + 6'SLN-CD | 0.051 | 0.306 | 0.436 | 0.385 | 0.068 | 0.452 |

The table shows p-values from within-condition comparisons of cell-type groups obtained with two different statistical methods both assuming the absence of differences as the null hypothesis. BB refers to the general linear hypothesis (two-sided) test on the coefficients from a Beta-Binomial model (see Methods). Chi2 refers to the Chi-squared contingency test on the counts of cells (in a specific condition) across compared cell types and viral load classes defined in Table 1 after merging Zero and Noise classes. The classes containing zero counts for both cell types were excluded in each comparison and the p-values were obtained by sampling the Chi-squared statistic under the null hypothesis i.e., the joint distribution being a (scaled) product of the marginal distributions.

52 host factor genes and retained those expressed in more than 50% of secretory and ciliated cells in (steady-state conditions) which resulted in 33 genes. We then tested for the differences in normalized expression between secretory and ciliated cells using Mann–Whitney–Wilcoxon test (with a two-sided alternative) and adjusted the resulting p-values using Bonferroni correction. Genes displayed in Supplementary Fig. 6 were selected manually, based on their relevance.

**Computational analysis of within-cell viral load.** First, we visualized and described within-cell frequencies of viral transcripts (FVTs) across groups using four discrete viral load classes introduced to better highlight the differences in viral load across individual cells, cell types, and conditions (Fig. 3 and Table 3). Next, we statistically investigated the differences across cell types and across conditions both in terms of (a) abundances of viral load classes, and, (b) continuous estimates of group-average FVTs from a Beta-Binomial regression model. Finally, we estimated and documented the global and cell-specific effects of treatments on FVT using data at our disposal and suggested the biological interpretation. Large proportions of uninfected (80–92%) and combined-treatment-infected cells (72–75%) contained exactly zero IAV transcripts and hence were assigned to Zero viral load class (Fig. 3 and Table 1). Similar to previous studies (75), we observed that small fractions of uninfected cells (10–20% depending on the cell type) had small but non-zero levels of viral load (FVTs below 0.02%). Since the distribution of FVTs in uninfected cells had some outliers at the right tail, we defined the right edge (left edge) of the Noise class (Low class) as the 99th percentile (0.012%) of FVTs in noninfected cells having at least one viral transcript. Except for 2% of ciliated cells, virtually all infected cells subject to the combined treatment (6'SLN-CD + IFN λ1 condition) were found to be below the maximum Noise level of 0.012%, making them hardly distinguishable from uninfected cells in terms of viral load (the results of statistical tests are reported later in this section). Every single one of the infected-untreated cells was found to be firmly above the maximum Noise level (0.012%) in terms of viral load. Indeed the minimum FVT across infected-untreated cells was found to be 0.066%, and that threshold was used to define the left edge (right edge) of Low (Medium) viral load classes. Since infected-untreated cells in all cell types except basal cells contained noticeable subpopulations with viral load well in excess of 0.066% (e.g., above 5%), we further divided the range of FVTs defining the High viral load class as containing the 25% of infected-untreated cells with the highest viral load (FVTs above 3.6%). These highly infected cells constituted 11%, 27%, and 55% of (infected untreated) BdiS, secretory and ciliated cells, respectively. For (infected untreated) ciliated cells, in fact, the most frequently observed viral load was close to 50% (the heavier right mode of a bimodal distribution).

The treated infected cells of all types had noticeably lower viral loads compared to the infected-untreated ones. Except for ciliated cells, the majority of infected cells treated with 6'SLN-CD were below the minimum FVT of untreated infected cells (0.066%), clearly highlighting the efficacy of the treatment. The IFN λ1 treatment resulted in further reduction of FVTs compared to 6'SLN-CD as evidenced by density plots and categorizations of cells by viral load classes shown in Fig. 3. In all cell types, the distributions of viral load classes were found to be different between 6'SLN-CD and IFN λ1 conditions as well as between individual treatments and the combined treatment (6'SLN-CD + IFN λ1) as reported below.

To assess differences in the viral load across conditions in a given cell type, we used both (1) a Chi-squared contingency test on class counts in condition pairs of interest after merging Noise and Zero viral load classes, and, (2) a test of a difference of estimated average FVTs being zero in a Beta-Binomial model fitted to each cell type—condition group (see Methods). For both methods and all cell types, the null hypotheses of no differences were strongly rejected (p-values below 1e-6) when (a) contrasting treatments versus each other in infected cells, (b) comparing combined treatment with either of the individual treatments in infected cells, (c) comparing each of the treatments with the untreated condition in infected cells. The null of no differences between (viral load in) combined-treatment condition and the uninfected-untreated cells was rejected only for ciliated cells (p-values of

0.004). For the rest of the cell types, the corresponding p-values were in excess of 0.14 (Chi-squared test) and. Finally, the viral loads of uninfected treated cells were found to be highly similar to these of uninfected-untreated cells (the nulls were not rejected with both testing methods with p-values exceeding 0.4.)

To examine the differences in viral load across cell types, we used similar methods. Since the average viral load in infected cells was found to be typically increasing across cell types ordered as basal, BdiS, secretory, and ciliated (Fig. 3 and Table 4), we tested for the differences in the same condition across subsequent pairs of cell types (Table 5).

In the infected-untreated condition, all differences were highly significant (all p-values below 2e-4) regardless of the testing method. For the infected cells treated with 6'SLN-CD the differences in viral load were similarly significant except for the one between BdiS and Secretory cells (p-value of 0.31 from a Beta-Binomial testing method)—in fact, the usual pattern of Secretory cells having higher average viral load compared to BdiS cells was reversed in that condition.

In the infected cells treated with IFN λ1, the differences between the (ordered pairs of) cell types were still highly significant according to the Beta-Binomial testing method (p-values below 0.001), while the Chi-squared test (strongly) rejected the null only when comparing the ciliated cells versus secretory cells (Table 3) highlighting the large differences between these cell groups in terms of their viral load classes (see Fig. 3). In the infected cells subject to the combined treatment the only significant difference was again the one between ciliated and secretory cells (p-values below 1e-4 for both testing method). Finally, the uninfected treated cells were generally indistinguishable in terms of viral load (when tested in the usual pairwise manner) regardless of the testing method, with the only exception being the borderline significant difference (p-value of 0.051 from Beta-Binomial method) between uninfected BdiS and basal cells.

**Human genome annotation.** GRCh38.p10 with only the main chromosome contigs retained i.e., chr1-chr22, chrX, chrY, and chrM.

**Human genome annotations.** Gencode release 29 with annotations of non-gene features (e.g., exons, transcripts, CDSs, and UTRs) removed if they overlapped with protein-coding or lincRNA features and did not have protein-coding, lincRNA, or processed-transcript tags themselves.

**Influenza A reference and annotations.** GCA_001343785.1 (https://www.ncbi.nlm.nih.gov/assembly/GCF_001343785.1). The viral reference annotations were preprocessed by prefixing all gene ids with Influenza A and by changing the type of CDS features to an exon.

**Statistics and reproducibility.** The number of replicates for each experiment is indicated in the legends of the corresponding figures. P-values for the drugs' antiviral effects were estimated by Student's t-test. Individual data points, the mean and standard deviation of the mean are shown for respective bar graphs. Single-cell data are based on a sequencing experiment performed on HAE obtained from a pool of donors as explained above, and the reported p-values were computed using the following methods: (a) differences in subcluster proportions across conditions—Chi-squared test; (b) differences in the fraction of viral transcripts across conditions—(two-sided) general linear test on coefficients of a Beta-Binomial model; (c) differences in abundances of viral load classes—Chi-squared test; (d) differences in the expression of the host factors between cell types—(two-sided) Mann–Whitney–Wilcoxon with Bonferroni correction. The statistical tests made on the single-cell data are included in the published code (see Code Availability) for reproducibility purposes.

**Reporting summary**. Further information on research design is available in the Nature Research Reporting Summary linked to this article.

## Data availability

The single-cell sequencing data for current study are available in the NCBI Gene Expression Omnibus repository under accession number GSE191176. The accession includes the raw FASTQ files and the unfiltered UMI count matrices in Hierarchical Data Format (HDF) produced by the Cell Ranger software by 10x Genomics. In addition, the matrices with the same format containing top 10,000 barcodes by UMI count are available at https://doi.org/10.5281/zenodo.7081937. The source data for main figures are available at https://doi.org/10.5281/zenodo.7082159.

## Code availability

The R code fully reproducing the analysis of single-cell sequencing data reported in the current study including the figures and the statistical tests is available at https://doi.org/10.5281/zenodo.7081572.

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

## Acknowledgements

We would like to thank the Genomic Platform at the University of Geneva) and Dr. Christel Borel for providing precious assistance and support in data analysis and experimental design. This work was supported by the Swiss National Science Foundation (Sinergia grant CRSII5_180323 to F.S. and C.T.) and by the Fondation Aclon (Geneva, to C.T.).

## Author contributions

C.M. and I.K. contributed equally to this work. C.M., I.X., E.T.D., and C.T. designed research. C.M., Y.Z., S.C., S.H., A.C.-A.Z., and V.C. performed research. C.M. and I.K. analyzed data. C.M. and C.T. wrote the paper. A.C.-A.Z., I.K., F.S., and I.X. read and edited the paper.

## Competing interests

The authors declare no competing interests.
