## [Peer Review File · Communications Biology]

Reviewers' comments:

Reviewer #1 (Remarks to the Author):

Here, Medaglia et al presented a study to evaluate the antiviral potential for influenza viruses using combination of two compounds, heptakis-(6-deoxy-6-thioundec)-beta-cyclodextrin (6'SLN-CD) and interferon lambda (IFN λ 1). They tested their antiviral efficacy in a 3D human airway epithelia (HAE) to represent human air-liquid interface. They used viral RNA quantities to represent virus content. They further interrogated the potential working mechanism through single cell RNA sequencing. They found there was a even less viral RNA from the samples treated with the combination of two compounds compared to the those from the samples treated with either one of those compounds. Thus, they concluded there is a synergistic effect with the treatment with combination of two compounds. Their single cell RNA sequencing provided initial observations on the potential action mode of the two compounds, i.e. which cell types were most affected.

Overall, this is a well written manuscript. However, there are major concerns that derailed my enthusiasm for this study.

1) There is lack of direct evaluation of virus concentrations. This could be easily achieved through a standard plaque assay on the biospecimen. Viral RNA copies have been shown to be disproportionately to virus concentrations. Therefore, I can not draw the similar conclusion with the data presented here.

2) There is practical concern based on the admiring regimen of their two compounds. In this study, they have tested a unrealistic administrative approach to deliver the two compounds. As detailed in their figure 1A, in order to achieve the combination therapy, they needed to treat cells with IFN λ 1 before the infection and 6'SLN-CD after infection. This regimen is nearly impossible to be applied to the real lives.

3) The mechanism study is still superficial. Their current data could not provide the vital connection step between these two seemingly unrelated compounds.

Minor Points;

"Current antiviral strategies are only efficient at the early stages of the infection and are challenged by the genomic instability of the virus." This is not accurate. One US approved anti-flu targets at NA protein to block the release of newly synthesized viral progenies.

Reviewer #2 (Remarks to the Author):

The paper by Medaglia et al is a clever approach to study IAV infection and replication in HAE cell models. It also describes the effect of a sialic acid decoy as well as IFN lambda. The data are fairly well described and will be resource for the field. Some issues need to be addressed, however.

1. I don't see a GEO or similar accession number for the scRNAseq data.
2. It's a bit surprising that the authors did not identify a Foxi1, Cfr expressing ionocyte cluster which is typically prominent in large airway and nasal airway per <https://www.covid19cellatlas.org/>. Also the authors should consider UMAP over TSNE.
3. Maybe I missed but I did not see where the authors assess IFN lambda receptors across the cell types and how this may explain the heterogeneous effects of IFN lambda.
4. In Figure 1 there is still a 4 log increase in viral RNA in the dual treatment group from 24 to 72 hours. How do the authors account for this?

Minor Comments:

The paper has changes tracked and does not appear to be in final submission form.

Reviewer #3 (Remarks to the Author):

What are the major claims of the paper?

The authors present a very interesting description of the anti influenza A virus (IAV) activity and transcriptomic impact of IFN lambda (IFN-L), 6'SLN-CD, and the combination of IFN-L/6'SLN-CD. The premise supporting selection of IFN-L as an anti-IAV host directed antiviral compound is previously described anti-influenza activity, similar to type I IFNs, but with smaller population of target cells than type I IFNs, which would be predicted to decrease toxicity. The premise supporting 6'SLN-CD as an anti-IAV compound comes from its direct acting antiviral activity trapping/inactivating newly formed virions, and prior studies by the authors ex vivo and in vivo (PMID: 33552848). Combination of these two antivirals, to harness both HDAC and DAA activity for complementary action, is unique. The use an infection model that combines a clinical IAV isolate (clinical A/Switzerland/3076/2016 H1N1 strain) not passaged in cell lines with a 3D primary human airway epithelial (HAE) air-liquid interface model. IFN-L was administered 24h before infection (HBI), in order to achieve antiviral effect through IFN effector activation prior to viral subversion (consistent with known requirements for IFN antiviral timing against other viruses such as dengue, PMID 28265266), at basal HAE side. 6'SLN-CD was dosed at 8 hours post infection. Antiviral effects of these two compounds were measured by 2 readouts 1) quantitation of virus RNA transcripts measured by single cell RNA seq, and 2) FACS. Number of IAV RNA copies measured from apical release was correlated with percent infected cells, as determined by FACS, identifying non-homogeneous infection (large number copies produced by a few cells). Antiviral effects were measured at 48h post infection to reach a well explained "fair compromise", as time point capable of capturing both antiviral and RNA modulating effect in order to detect log phase viral replication prior to virus-mediated shutoff of host transcription or cytotoxicity.

The authors find that treatment with both IFN-L and 6'SLN-CD alone suppresses IAV, but antiviral suppression is greatest with combined treatment, which is minimally toxic.

They also, using scRNAseq, find that basal, secretory, ciliated cells are not affected by IAV or antiviral treatments, and identify observed cell types and proportions are consistent with prior scRNAseq studies supporting similarity between their ex vivo model and in vivo respiratory epithelium. Interestingly, at 48h neither infection alone nor treatment led to >2x change in relative proportion of cell types. Quantity of infection per cell type was measured as zero (Z), noise (N), low (L), medium (M), high (H). Secretory vs ciliated cells were found to have higher expression of unique dependency factors. 6'SLN-CD blocked IAV in secretory > ciliated cells, whereas IFN and 6'SLN showed equal IAV suppression in secretory and ciliated cells. Consistently, ISG activation was similar across cell types. Basal cells were exhibited fewest virus transcripts, clinically important due to their role maintaining the barrier of respiratory epithelium. Interestingly, virus RNA production was not uniform across mRNA segments, with ratios consistent with prior reports.

Are they novel and will they be of interest to others in the community and the wider field?

--- Yes, these claims are novel and will be of interest to others in the virology community and the broader clinical/public health community. Neither IFN-L nor 6'SLN-CD are standard of care treatments for IAV.

--- While a recurring critique of IFN as an antiviral is the need to pre-administer, it is important to note that in a crisis targeted pre-administration may actually be more feasible than accommodating multiple patients in need of hospitalization. Furthermore, a requirement for pre-administration ex vivo may still be clinically useful for those who are nasal swab positive but early in infections.

--- scRNAseq results are interesting and well presented, showing different cell type anti-IAV activity of 6'SLN-CD vs IFN-L, which may help future targeting / drug delivery strategies

If the conclusions are not original, it would be helpful if you could provide relevant references.

--- n.a.

Is the work convincing, and if not, what further evidence would be required to strengthen the conclusions?

--- In the introduction, please change "investigated mechanism of action" to "investigated transcriptomic impact" to more accurately reflect the data presented.

--- If it would be possible to test HAE cell lysate or supernatant w/ w/o combination treatment for ability to inhibit influenza plaques, seeking to determine inhibition of live virus by IFN-L and 6'SLN-CD, this would provide further evidence of antiviral strength of this compound. If this has already been performed, please provide reference. If not feasible to set up these experiments, at least discuss relevance of plaque assays in the text (perhaps as a future direction for more lifecycle/antiviral impact information on these compounds).

---virus qRT PCR: please include the primers used to quantify IAV (rather than a reference to another publication)

---analysis of host factors, methods section: please describe how the authors "constructed a list of 52 host factors"

---please comment on how single cell frequency of viral transcript (FVT) data can be interpreted for clinical relevance: what level of infection, observed in untreated cells, and log-fold decrease, or percent infected cell decrease, would the authors consider adequate for moving a drug/compound toward clinical trials ? How can this evolving field of FVTs be compared to known antiviral compounds / existing data sets?

On a more subjective note, do you feel that the paper will influence thinking in the field?

--- Yes, there is a major clinical need for additional anti-IAV countermeasures. A major IAV pandemic is highly likely, and we are fully unprepared. This work provides rationale for further clinical study of IFN-L / 6'SLN-CD.

Please feel free to raise any further questions and concerns about the paper.

--- None.

Appropriateness and validity of any statistical analysis, as well the ability of a researcher to reproduce the work, given the level of detail provided.

--- Appropriate and valid, and I believe enough detail was provided for reproduction other than need to include the qRT PCR IAV primers, not a reference to another manuscript.

We would like to thank the reviewers for their careful reading of the manuscript and their constructive and valuable comments, which we have implemented in the revised version. Please find below a point-by-point response to all comments.

Reviewers' comments

Reviewer #1 (Remarks to the Author):

Here, Medaglia et al presented a story to evaluate the antiviral potential for influenza viruses using combination of two compounds, heptakis-(6-deoxy-6-thioundec)-beta-cyclodextrin (6'SLN-CD) and interferon lambda (IFN λ 1). They tested their antiviral efficacy in a 3D human airway epithelia (HAE) to represent human air-liquid interface. They used viral RNA quantities to represent virus content. They further interrogated the potential working mechanism through single cell RNA sequencing. They found there was a even less viral RNA from the samples treated with the combination of two compounds compared to the those from the samples treated with either one of those compounds. Thus, they concluded there is a synergistic effect with the treatment with combination of two compounds. Their single cell RNA sequencing provided initial observations on the potential action mode of the two compounds, i.e. which cell types were most affected.

Overall, this is a well written manuscript. However, there are major concerns that derailed my enthusiasm for this story.

1) There is lack of direct evaluation of virus concentrations. This could be easily achieved through a standard plaque assay on the biospecimen. Viral RNA copies have been shown to be disproportionally to virus concentrations. Therefore, I can not draw the similar conclusion with the data presented here.

We thank the reviewer for his/her comment. Indeed, viral RNA copies may not give an exact indication on the number of infectious viral particles, as they also consider defective viral particles. To address this point, we titrated by plaque assay the apical wash of the tissue used to perform the RT-qPCR of figure 2. We observed that the plaque assay measurements were consistent with the RT-qPCR ones. The results reported **Reviewer Figure 1B**, included in the revised version of the manuscript (Fig. S1D new panel B and methods section lines 578-583) confirm the observations made at the RNA level.

Figure 1. Combinatorial effect of 6'SLN-CD + IFN λ 1 in HAE. **A)** Bar plot showing the kinetic of IAV replication in HAE treated with 6'SLN-CD only, or with IFN λ 1 only, or with both compounds. **B)** Titration, by plaque assay in MDCK cells, of the HAE apical washes collected at 48 hours post infection. The results represent three independent experiments conducted in duplicate in HAE developed from a pool of donors and infected with 10^3 RNA copies of clinical A/Switzerland/3076/2016 H1N1 (0 h corresponds to the time of viral inoculation). Viral replication in A was assessed measuring the apical release of IAV by RT-qPCR. *, $p \leq 0.05$; **, $p \leq 0.01$; ***, $p \leq 0.001$; ****, $p \leq 0.0001$.

2) There is practical concern based on the admiring regimen of their two compounds. In this study, they have tested a unrealistic administrative approach to deliver the two compounds. As detailed in their figure 1A, in order to achieve the combination therapy, they needed to treat cells with IFN λ 1 before the infection and 6'SLN-CD after infection. This regimen is nearly impossible to be applied to the real lives.

- We appreciate the reviewer's observation. We know that IFN λ pre-treatment is not an attainable clinical option, but both in cell lines and human airway epithelia this cytokine exerts an antiviral effect only when administered in pre- or in both pre- and post-treatment. This statement is supported by several publications (N. A. Ilyushina et al., *PLoS One* 2017; K. Minton, *Nat Rev Immunol* 2017) as well as by a number of unpublished data (not included in the revised version of the manuscript), shown in **Reviewer Figure 2**.

-More specifically, *in vivo*, IFN λ is sensed by the trans-epithelial dendritic cells of the respiratory mucosa, which strongly amplify its signal (I. Latino et al., *Current Opinion in Physiology* 2021).

-The ultimate goal of our work is to demonstrate the beneficial effect of IFN λ 1 and 6'SLN-CD in a relevant *ex vivo* model, based on the evidence that in *in vivo* models of influenza IFN λ inhibits viral replication when administered intranasally after the infection.

-Lastly, in an emergency action plan, targeted pre-administration may be clinically useful for those who are nasal swab positive but still in the early stages of infection, or at-risk persons in contact with infected individuals who could take the treatment in prevention.

- To address the reviewer's concern, we better clarify this point in the discussion section stating that

“Additional investigations in more relevant in vivo models of infection, such as mice or ferrets, will be necessary to further assess the efficacy of IFN λ 1 + 6'SLN-CD formulation, with a more realistic administration protocol where both compounds are administered in post-treatment, and its genetic barrier to antiviral resistance.”

Reviewer Figure 2. A. Antiviral activity of hIFN λ 1 in Calu-3 cells against H1N1pdm09. Cells were infected with a MOI of 0.1 PFU/cell. At 1 hpi the inoculum was removed and the cells were overlaid with a dose-range of drug in serum-free DMEM. At 24 hpi the infectious supernatant was collected and the viral titer was determined by plaque assay in MDCK cells. B. Determination of hIFN λ 1 EC50 in Calu-3 cells against H1N1pdm09, relatively to panel A. C. Determination of hIFN λ 1 EC50 in A549 cells against H1N1pdm09, with the same treatment protocol described in A. Unlike Calu-3, A549 cells were infected with a MOI of 0.5 and the drug dose-range was dissolved in DMEM + TRCK trypsin. The percentages of infected cells were determined, compared to an untreated control, at 24 hpi by ICC. D. Antiviral activity of hIFN λ 1 (dose) in human reconstituted airway epithelia. HAE were infected with 1E4 RNA copies of clinical H1N1 A/Switzerland/3076/16 strain. IFN λ 1 (10ng/ml) was administered daily at 24 hbi (pre-treatment) or at 8 hpi (post-treatment). Viral replication was assessed measuring the apical release of IAV by RT-qPCR. The results of two independent experiments conducted in duplicate are shown.

3) The mechanism study is still superficial. Their current data could not provide the vital connection step between these two seemingly unrelated compounds.

We agree with the reviewer's point. Mechanistically, the link between the two compounds is not explained. Interestingly, we highlight that the single and combined treatments inhibit viral replication to different extents in different cell types, although additional investigations are needed to better explain this observation. It seems thus that as expected, IFN λ 1 and 6'SLN-CD are not linked, they target IV from different angles, without intersecting. This is the reason why we chose to combine them, as they complement each other and comply with the rationale of antiviral combined therapies.

As far as we know, IFN λ 1 does not affect sialic acid cluster size, thus not trespassing on the range of action of 6'SLN-CD, and vice versa. Obviously, and as stated in the discussion, more *in vivo* experiments must be performed to better clarify the mechanism of action of the combined treatment.

Minor Points;

“Current antiviral strategies are only efficient at the early stages of the infection and are challenged by the genomic instability of the virus.” This is not accurate. One US approved anti-flu targets NA protein to block the release of newly synthesized viral progenies.

We thank the reviewer for raising this point, as it highlights a misleading sentence. By “early stages of the infection” we did not mean “early stages of viral replication”, but simply a short time after the onset of the symptoms. Oseltamivir, that inhibits the very last step of viral cycle, has a beneficial effect only if assumed within 48 hours after onset of flulike symptoms.

To address the reviewer's concern, we corrected our statement as follows “Current antiviral strategies are only efficient when administered within a short time after the onset of the symptoms and are challenged by the genomic instability of the

virus” (lines 446 - 448) in the revised version of the manuscript.

Reviewer #2 (Remarks to the Author):

The paper by Medaglia et al is a clever approach to study IAV infection and replication in HAE cell models. It also describes the effect of a sialic acid decoy as well as IFN lambda. The data are fairly well described and will be resource for the field. Some issues need to be addressed, however.

1. I don't see a GEO or similar accession number for the scRNAseq data.

The GEO accession number is [GSM5740432](https://www.ncbi.nlm.nih.gov/geo/query/acc.cgi?acc=GSM5740432) (methods section line 660).

We apologize for not including the submission ID in the manuscript. The data had been transferred to GEO by the time we submitted the manuscript but the GEO identifier had not yet been assigned (or communicated to us).

2. It's a bit surprising that the authors did not identify a Foxi1, Cfr expressing ionocyte cluster which is typically prominent in large airway and nasal airway per <https://www.covid19cellatlas.org/>. Also the authors should consider UMAP over TSNE.

- To clarify the possibility of detection of the ionocyte cluster we first assessed to what extent the FOXI1 and CFTR genes are expressed in our data. **Reviewer Figure 3** demonstrates the histograms of these genes (along with the marker genes for the cell types identified in the paper and the IFN lambda receptor genes discussed below).

Overall, the expression levels for these two genes in our data are very low: in particular, FOXI1 is expressed in only 34 cells (0.3% of total) while CFTR is expressed in 604 cells (roughly 5%).

We have also highlighted the cells expressing FOXI1 and the cells expressing CFTR on a t-SNE plot (**Reviewer Figure 4**) and did not observe clear clustering patterns for these cells.

Reviewer Figure 3. Histograms of UMI counts of marker genes, ionocyte markers and interferon lambda receptors. The panel on the right are zoomed-in views or regions marked with green rectangles on the left panel.

- UMAP projections were used along with the t-SNE in the early versions of the paper. The clustering patterns were quite similar and we did not observe UMAP to be beneficial for presenting our findings or more “meaningful” otherwise (in line with Kobak D. and Linderman G.C. Nat Biotechnol 2021). We settled on keeping t-SNE projections as this method is slightly older and hence is likely to be familiar to a wider audience.

Reviewer Figure 4. Expression of ionocyte markers and interferon lambda receptors in t-SNE coordinates.

3. Maybe I missed but I did not see where the authors assess IFN lambda receptors across the cell types and how this may explain the heterogeneous effects of IFN lambda.

- We agree that the assessment of IFN lambda receptors would be a desirable addition to the paper and performed the analysis, as requested by the reviewer. We observed low expression levels of both IFNLR1 and IL10RB genes in our data (**Reviewer Figure 3**). Indeed, there are no cells with more than 3 UMIs for either gene and the total number of cells where these genes are expressed are 656 and 892 respectively (around 5 and 7% of all barcodes).

- To perform the differential expression analysis for the IFNLR1 gene we employed a methodology similar to the assessment of the viral load. More specifically, we used a Beta-Binomial regression model fit to IFNLR1 and non-IFNLR1 UMI counts and assessed general linear hypotheses a) that IFNLR1 load being is independent of conditions (within the same cell type), and b) that IFNLR1 load being is independent of the cell type (within the same condition).

- We observed that the IFNLR1 expression in all conditions on average higher in Ciliated cells compared to all other cell types (**Reviewer Table 1**). These differences are statistically significant for all conditions except infected untreated (**Reviewer Table 2**).

- Upon contrasting the expressions of IFNLR1 gene across conditions within the same cell types we found that the infected cells treated only with interferon lambda have (on average) same or higher expressions of IFNLR1 gene compared to infected untreated cells (with differences being highly statistically significant for Ciliated cells) and compared to cyclodextrin-treated cells (with differences being statistically significant for all cell types except BdiS and highly significant for Ciliated cells).

- We furthermore observed that double-treated infected cells have generally similar IFNLR1 expressions to interferon lambda-treated infected cells and that uninfected double-treated cells are generally comparable to non-infected untreated cells in terms of IFNLR1 expression (**Reviewer Table 1 and 2**). Finally, we observed a slight decrease in IFNLR1 expression in infected untreated cells compared to steady state for Basal and Ciliated cells (but not the other cell types) and these differences were not statistically significant (**Reviewer Table 3**).

Reviewer Table 1. Average within-cell frequencies of IFNLR1 transcripts across cell types and conditions				
Condition	Basal	BdiS	Secretory	Ciliated
Mock steady state	2.4ppm (1.5ppm-4.0ppm)	1.2ppm (0.4ppm-3.4ppm)	1.9ppm (1.0ppm-3.5ppm)	5.6ppm (3.7ppm-8.6ppm)
Infected untreated	1.9ppm (1.2ppm-3.0ppm)	1.7ppm (0.7ppm-4.3ppm)	2.5ppm (1.3ppm-4.8ppm)	3.7ppm (2.3ppm-6.1ppm)
Infected + 6'SLN-CD	1.2ppm (0.8ppm-2.1ppm)	1.7ppm (0.9ppm-3.1ppm)	1.1ppm (0.5ppm-2.1ppm)	3.1ppm (1.9ppm-4.9ppm)
Infected + IFN λ1	2.2ppm (1.4ppm-3.5ppm)	2.7ppm (1.4ppm-5.1ppm)	2.5ppm (1.5ppm-4.1ppm)	6.8ppm (4.8ppm-9.7ppm)
Infected + IFN λ1 + 6'SLN-CD	2.0ppm (1.4ppm-3.0ppm)	1.2ppm (0.5ppm-3.0ppm)	2.6ppm (1.6ppm-4.2ppm)	6.8ppm (5.0ppm-9.2ppm)
Mock + IFN λ1 + 6'SLN-CD	1.4ppm (0.9ppm-2.2ppm)	1.0ppm (0.4ppm-2.2ppm)	1.1ppm (0.6ppm-2.1ppm)	4.2ppm (2.8ppm-6.3ppm)

Reviewer Table 1. The table shows the point estimates (as parts per million) and the 99% confidence intervals from a Beta-Binomial regression model. The estimates are naturally on log odds scale and were converted to probabilities for presentation. Note that the difference of parameter estimates is not necessarily insignificant if the respective confidence intervals overlap – such an assessment is overly conservative

Condition	BdiS vs Basal	Secretory vs BdiS	Ciliated vs Secretory
Mock steady state	0.106	0.321	<0.001
Infected untreated	0.813	0.425	0.204
Infected + 6'SLN-CD	0.340	0.200	<0.001
Infected + IFN λ 1	0.543	0.788	<0.001
Infected + IFN λ 1 + 6'SLN-CD	0.173	0.062	<0.001
Mock + IFN λ 1 + 6'SLN-CD	0.275	0.658	<0.001

Reviewer Table 2. Shown are p-values from within-condition comparisons of cell type groups obtained with a general linear hypothesis (two-sided) test on the coefficients from a Beta-Binomial model. The null hypotheses are that the distributions of IFNLR1 UMI fractions are independent from the cell types.

Contrasted Conditions		Cell Type			
Condition A	Condition B	Basal	BdiS	Secretory	Ciliated
Infected untreated	Mock steady state	0.348	0.467	0.448	0.098
Infected + IFN λ 1	Infected untreated	0.518	0.313	>0.999	0.009
Infected + 6'SLN-CD	Infected + IFN λ 1	0.025	0.175	0.011	<0.001
Infected + 6'SLN-CD	Infected untreated	0.105	0.933	0.023	0.483
Infected + IFN λ 1 + 6'SLN-CD	Infected + IFN λ 1	0.690	0.065	0.887	0.946
Infected + IFN λ 1 + 6'SLN-CD	Infected + 6'SLN-CD	0.044	0.447	0.007	<0.001
Infected + IFN λ 1 + 6'SLN-CD	Infected untreated	0.766	0.468	0.904	0.008
Infected + IFN λ 1 + 6'SLN-CD	Mock steady state	0.469	0.957	0.316	0.363
Mock + IFN λ 1 + 6'SLN-CD	Mock steady state	0.041	0.703	0.152	0.196

Reviewer Table 3. Shown are p-values from within-cell-type comparisons of condition groups obtained with a general linear hypothesis (two-sided) test on the coefficients from a Beta-Binomial model. The null hypothesis is that the distributions of IFNLR1 UMI fractions are independent from the conditions.

4. In Figure 1 there is still a 4 log increase in viral RNA in the dual treatment group from 24 to 72 hours. How do the authors account for this?

We wanted to assess the effect of the treatments on both the host and the virus, therefore we selected a dosage of drugs able to significantly inhibit viral replication without eradicating the virus. This dosage of treatments was kept constant over time, despite viral replication, which is exponential. Considering that our human airway epithelia lack immune cells an increase of viral RNA (by 2 log) from 48 and 72 hpi is expected.

Minor Comments:

The paper has changes tracked and does not appear to be in final submission form.

We properly formatted the manuscript.

Reviewer #3 (Remarks to the Author):

What are the major claims of the paper?

The authors present a very interesting description of the anti influenza A virus (IAV) activity and transcriptomic impact of IFN lambda (IFN-L), 6'SLN-CD, and the combination of IFN-L/6'SLN-CD. The premise supporting selection of IFN-L as an anti-IAV host directed antiviral compound is previously described anti-influenza activity, similar to type I IFNs, but with smaller population of target cells than type I IFNs, which would be predicted to decrease toxicity. The premise supporting 6'SLN-CD as an anti-IAV compound comes from its direct acting antiviral activity trapping/inactivating newly formed virions, and prior studies by the authors *ex vivo* and *in vivo* (PMID: 33552848). Combination of these two antivirals, to harness both HDAC and DAA activity for complementary action, is unique.

The use an infection model that combines a clinical IAV isolate (clinical A/Switzerland/3076/2016 H1N1 strain) not

passed in cell lines with a 3D primary human airway epithelial (HAE) air-liquid interface model. IFN-L was administered 24h before infection (HBI), in order to achieve antiviral effect through IFN effector activation prior to viral subversion (consistent with known requirements for IFN antiviral timing against other viruses such as dengue, PMID 28265266), at basal HAE side. 6'SLN-CD was dosed at 8 hours post infection. Antiviral effects of these two compounds were measured by 2 readouts 1) quantitation of virus RNA transcripts measured by single cell RNA seq, and 2) FACS. Number of IAV RNA copies measured from apical release was correlated with percent infected cells, as determined by FACS, identifying non-homogeneous infection (large number copies produced by a few cells). Antiviral effects were measured at 48h post infection to reach a well explained "fair compromise", as time point capable of capturing both antiviral and RNA modulating effect in order to detect log phase viral replication prior to virus-mediated shutoff of host transcription or cytotoxicity.

The authors find that treatment with both IFN-L and 6'SLN-CD alone suppresses IAV, but antiviral suppression is greatest with combined treatment, which is minimally toxic.

They also, using scRNAseq, find that basal, secretory, ciliated cells are not affected by IAV or antiviral treatments, and identify observed cell types and proportions are consistent with prior scRNAseq studies supporting similarity between their *ex vivo* model and *in vivo* respiratory epithelium. Interestingly, at 48h neither infection alone nor treatment led to >2x change in relative proportion of cell types. Quantity of infection per cell type was measured as zero (Z), noise (N), low (L), medium (M), high (H). Secretory vs ciliated cells were found to have higher expression of unique dependency factors. 6'SLN-CD blocked IAV in secretory > ciliated cells, whereas IFN and 6'SLN showed equal IAV suppression in secretory and ciliated cells. Consistently, ISG activation was similar across cell types. Basal cells were exhibited fewest virus transcripts, clinically important due to their role maintaining the barrier of respiratory epithelium. Interestingly, virus RNA production was not uniform across mRNA segments, with ratios consistent with prior reports.

Are they novel and will they be of interest to others in the community and the wider field?

--- Yes, these claims are novel and will be of interest to others in the virology community and the broader clinical/public health community. Neither IFN-L nor 6'SLN-CD are standard of care treatments for IAV.

--- While a recurring critique of IFN as an antiviral is the need to pre-administer, it is important to note that in a crisis targeted pre-administration may actually be more feasible than accomodating multiple patients in need of hospitalization. Furthermore, a requirement for pre-administration *ex vivo* may still be clinically useful for those who are nasal swab positive but early in infections.

--- scRNAseq results are interesting and well presented, showing different cell type anti-IAV activity of 6'SLN-CD vs IFN-L, which may help future targeting / drug delivery strategies

If the conclusions are not original, it would be helpful if you could provide relevant references.

--- n.a.

Is the work convincing, and if not, what further evidence would be required to strengthen the conclusions?

--- In the introduction, please change "investigated mechanism of action" to "investigated transcriptomic impact" to more accurately reflect the data presented.

We thank the reviewer for his/her comment. We modified the text accordingly.

--- If it would be possible to test HAE cell lysate or supernatant w/ w/o combination treatment for ability to inhibit influenza plaques, seeking to determine inhibition of live virus by IFN-L and 6'SLN-CD, this would provide further evidence of antiviral strength of this compound. If this has already been performed, please provide reference. If not feasible to set up these experiments, at least discuss relevance of plaque assays in the text (perhaps as a future direction for more lifecycle/antiviral impact information on these compounds).

We addressed the reviewer's concern. Please see the answer to the first question of reviewer 1 (page 1).

---virus qRT PCR: please include the primers used to quantify IAV (rather than a reference to another publication)

We included the sequence of the primers and the probes used to quantify IAV in the methods of the revised version of the manuscript (Table 1).

---analysis of host factors, methods section: please describe how the authors "constructed a list of 52 host factors"

The list of 52 host factors was constructed making an extensive search in the literature (Ref. 66 - 70 in the manuscript).

---please comment on how single cell frequency of viral transcript (FVT) data can be interpreted for clinical relevance: what level of infection, observed in untreated cells, and log-fold decrease, or percent infected cell decrease, would the authors consider adequate for moving a drug/compound toward clinical trials?

How can this evolving field of FVTs be compared to known antiviral compounds / existing data sets?

We thank the reviewer for this comment, as it allows us to further highlight the relevance of our study. Measuring the frequency of viral transcripts (i.e. % of viral transcripts in the cell) could be used as a proxy for the evaluation of compounds that inhibit viral replication intracellularly in each different cell type and to, as we showed in our work, stratify the antiviral effect based on the heterogeneous range of action of the drug at the tissue level. ScRNA-seq is a relatively new methodology, still quite expensive, but its costs are diminishing over time. We envision that in the future scRNA-seq

analysis will be used not only in cell lines, tissue culture models but also in antiviral clinical trials, for example in respiratory cells collected from nasal swabs, to measure the above-mentioned parameters, to compare the effect of multiple antivirals in human patients, based on pre/existing datasets and lastly to better address the differences in antiviral efficacy between different individuals. The availability of such data will help to decide, based on preclinical testing, which antiviral represent a promising compound for further *in vivo* studies. The latter aspect is of high importance, in the era of precision medicine.

On a more subjective note, do you feel that the paper will influence thinking in the field?

--- Yes, there is a major clinical need for additional anti-IAV countermeasures. A major IAV pandemic is highly likely, and we are fully unprepared. This work provides rationale for further clinical study of IFN-L / 6'SLN-CD.

Please feel free to raise any further questions and concerns about the paper.

--- None.

Appropriateness and validity of any statistical analysis, as well the ability of a researcher to reproduce the work, given the level of detail provided.

--- Appropriate and valid, and I believe enough detail was provided for reproduction other than need to include the qRT PCR IAV primers, not a reference to another manuscript.

Reviewers' comments:

Reviewer #1 (Remarks to the Author):

The modified manuscript has addressed my main concern.

Reviewer #2 (Remarks to the Author):

The authors have largely addressed my concerns. However, the fact that the authors could not identify ionocytes in their scRNAseq data is concerning. This does not appear to be due to cell number as the number of cells captured is sufficient to detect ionocytes. This may be due to too shallow of sequencing the library. The authors mention filtering on UMIs but they should also present the number of mapped reads per cell after sequencing the library to assure there was adequate coverage. Saturation plots are part of the Cell Ranger QC metric.

Reviewer #3 (Remarks to the Author):

The authors have addressed my questions. I feel this manuscript is ready for publication.

We are grateful to the reviewers for their work, constructive and valuable comments. The concerns raised by the Reviewer #2 are addressed below.

Reviewer #2 (Remarks to the Author):

The authors have largely addressed my concerns. However, the fact that the authors could not identify ionocytes in their scRNAseq data is concerning. This does not appear to be due to cell number as the number of cells captured is sufficient to detect ionocytes.

We respectfully disagree with these statements. The estimated relative frequency of ionocytes is only 0.03% (3 cells per 10,000 on average) in the human lung cell atlas dataset [1, Table S1]. Hence it is perfectly normal we did not detect a ionocyte cluster within the set of 12,778 cells selected for the analysis in our study.

This may be due to too shallow of sequencing the library. The authors mention filtering on UMIs but they should also present the number of mapped reads per cell after sequencing the library to assure there was adequate coverage. Saturation plots are part of the Cell Ranger QC metric.

As noted above, we believe that non-detection of ionocytes is the consequence of rarity of such cells (in relation to the number of analyzed cells). Hence while it is plausible that shallow sequencing might result in non-detection of certain cell types, we do not see it as a possible reason in our case.

We agree, nonetheless, that a brief description of some of the quality metrics produced by Cell Ranger would be appropriate in the manuscript and included such discussion in the Materials and Method Section. The key metrics are also presented in the Table R2.1 on page 2.

The average number of reads per cell is above 65,000 for all samples and exceeds 100,000 for all samples except one. These numbers are comparable to those reported for some of human cell atlas datasets (for instance, 90,000 and 40,000 reads per cell on average in [2] and [3] respectively) and are above what is typically considered to be a shallow single-cell sequencing (e.g. 14,000 raw reads per cell on average in [4]).

Please note that the median number of genes per cell in our dataset is also comparable to those reported for the abovementioned human cell datasets (4,000-5,200 in our study vs 2,000 in [2]) and is much higher than in a typical shallow sequencing experiment (e.g. 524 median genes per cell in [4]).

References

- [1] Travaglini KJ et al. A molecular cell atlas of the human lung from single-cell RNA sequencing. Nature. 2020 Nov; 587(7835):619-625. <https://pubmed.ncbi.nlm.nih.gov/33208946/>
- [2] Muraro MJ et al. A Single-Cell Transcriptome Atlas of the Human Pancreas. Cell Syst. 2016 Oct 26;3(4):385-394.e3. <https://www.ncbi.nlm.nih.gov/pmc/articles/PMC5092539/>
- [3] Lukowski SW et al. A single-cell transcriptome atlas of the adult human retina. EMBO J. 2019 Sep 16;38(18):e100811. <https://www.ncbi.nlm.nih.gov/pmc/articles/PMC6745503/>
- [4] Cao J et al. A human cell atlas of fetal gene expression. Science. 2020 Nov 13;370(6518):eaba7721. <https://www.ncbi.nlm.nih.gov/pmc/articles/PMC7780123/>

Tables

Sample	Estimated Number of Cells	Mean Reads per Cell	Median Genes per Cell	Median UMIs per Cell	Mapped Confidently to Genome	Mapped Confidently to Transcripts
Infected untreated	1649	135968	4624	25810	87.2%	63.90%
Infected cyclodextrin	3574	67820	3998	17031	88.3%	65.40%
Infected interferon lambda	2036	117047	4631	20792	85.3%	61.20%
Infected combined treatment	2201	109411	5081	24934	86.9%	60.60%
Uninfected untreated	2032	100124	4326	18328	87.2%	63.10%
Uninfected combined treatment	2313	101336	5233	27419	87.3%	63.00%

Table R2.1 Key quality metrics from Cell Ranger. Please note that the metrics were computed on 13,805 cells selected to have more than 10,000 distinct UMIs – a slightly larger set than 12,778 cells included in the analysis because the cells including more than 15% of mitochondrial UMIs were not yet excluded.